# Children's family income is associated with cognitive function and volume of anterior not posterior hippocampus

Alexandra L. Decker [1✉], Katherine Duncan[1,4], Amy S. Finn[1,4] & Donald J. Mabbott [1,2,3,4]

Children from lower income backgrounds tend to have poorer memory and language abilities than their wealthier peers. It has been proposed that these cognitive gaps reflect the effects of income-related stress on hippocampal structure, but the empirical evidence for this relationship has not been clear. Here, we examine how family income gaps in cognition relate to the anterior hippocampus, given its high sensitivity to stress, versus the posterior hippocampus. We find that anterior (but not posterior) hippocampal volumes positively correlate with family income up to an annual income of ~$75,000. Income-related differences in the anterior (but not posterior) hippocampus also predicted the strength of the gaps in memory and language. These findings add anatomical specificity to current theories by suggesting a stronger relationship between family income and anterior than posterior hippocampal volumes and offer a potential mechanism through which children from different income homes differ cognitively.

[1] Department of Psychology, University of Toronto, Toronto, ON, Canada. [2] Neurosciences and Mental Health, Hospital for Sick Children, Toronto, ON, Canada. [3] Department of Psychology, Hospital for Sick Children, Toronto, ON, Canada. [4]These authors contributed equally: Katherine Duncan, Amy S. Finn, Donald J. Mabbott. ✉email: Alexandra.decker@mail.utoronto.ca

As early as primary school, children with lower socio-economic status (SES) fall up to a full standard deviation behind their higher income peers on measures of episodic memory and language abilities on average[1,2]. These cognitive gaps coincide with differences in brain structure that are thought to reflect the cumulative effects of low-income-related stress[3–6] and high-income-related environmental stimulation on brain development[3,7]. Indeed, being part of a lower income household is associated with possessing fewer material and nonmaterial resources[3,7] and incurring a higher number of stressful life events[4,8–11], with links between income and stress persisting up to an annual income of ~$75,000–$95,000[8,9]. One brain region—the hippocampus—has been singled out by prominent theories to explain income-gaps in cognition[3,5–7,12], because it is particularly sensitive to chronic stress[4,13–17], is smaller in children with low SES[2–4,18], and is important for both episodic memory[19] and aspects of language[20–24]. With this trifecta of findings, it is tempting to conclude that memory and language gaps in lower versus higher income children partially reflect structural differences in the hippocampus. When tested empirically, however, income differences in hippocampal structure seemingly do not mediate these cognitive gaps[4,18,25].

Here, we identify a critical factor that may have previously obscured the role of the hippocampus in income-related gaps in memory and language; prior research has treated the hippocampus as a homogeneous structure[2,4,18], but mounting evidence indicates that the hippocampus is, in fact, comprised of distinct anterior and posterior regions. These distinct regions have different developmental trajectories[26], different roles in cognition[27], and—importantly—different vulnerabilities to chronic stress. Indeed, work in both humans and animals shows that stress preferentially impacts the anterior hippocampus[14,15,28–31]. Stressed animals display disproportionate decreases in cell survival and neurogenesis in the ventral (anterior) portion of the hippocampus[28,30–32], and in humans[14,15] and nonhuman primates[29], stress is linked to smaller anterior hippocampal volumes. Furthermore, antidepressants—a common treatment for stress-related disorders—selectively increase neurogenesis in the *anterior* hippocampus[28,32]. Likewise, longitudinal work in humans shows that resolving pathological over-secretions of stress hormones selectively increases anterior, but not posterior hippocampal volumes[16]. The anterior (rodent ventral) hippocampus is also involved in regulating stress and anxiety-related behaviors[33], and physiological responses to stress[34] via dense structural connectivity with the amygdala[35], and hypothalamic nuclei[27]. Together, this work suggests that children growing up in lower income homes, who are more likely to experience stressful events than their higher income peers[4,36–38], may have disproportionately smaller anterior hippocampi.

We therefore investigate the association between family income and regional hippocampal volumes in a large pre-existing dataset that included data from children, adolescents, and young adults ($n = 703$; mean age = 12.3, range = 3–21). We also examine the relationship between regional hippocampal volumes and income-related differences in cognitive scores and perform exploratory analyses to test whether age moderates the results. Given the link between stress and lower income[8,9], we hypothesize that low income might disproportionately influence the size of the anterior (rather than posterior) hippocampus. We expect to observe this relationship up to an annual income of ~75k annually because stress positively correlates with income up until this threshold[8,9]. Moreover, if this is the case, we predict that the anterior hippocampus mediates income-related gaps in cognitive abilities that are supported by the hippocampus (i.e., memory and vocabulary). Importantly, although our hypotheses are inspired by animal work on the impact of chronic stress on the anterior hippocampus, we do not have a direct measure of stress in our dataset

and are therefore unable to directly test the impact of stress specifically.

All participants underwent magnetic resonance (MR) scanning and 690 participants in the sample completed assessments of memory and vocabulary. We include episodic associative memory and vocabulary assessments in our investigation because both memory[39,40] and the acquisition and use of new vocabulary[20,21,23,24,41–43] are thought to depend on hippocampal binding. Individuals with hippocampal damage are impaired at learning the meaning of new words[42,44], and studies demonstrate that language acquisition and vocabulary correlate with hippocampal activity[21,24], and hippocampal volume in children[45] and adults[46]. To test the specificity of the relationships between hippocampal volumes and cognition, we include performance on a processing speed task[47] that is independent of the hippocampus[39].

Family income was reported by accompanying parents or, in some cases, young adult participants (range = <$5k to >$300k annually). Of note, the log transformation of family income is used in the analyses to reflect the idea that adding a given income increment to a lower income family has more impact than the same increment to a higher income family. Thus, using the log of income reflects the greater impact that income increments have on lower than higher income individuals. We directly test this widely held assumption in our data by examining whether linear gains in income correspond to larger increases in cognitive scores and hippocampal volumes among individuals from lower (≤$75k) than higher income families (>$75k). While these analyses are exploratory, we expect that linear gains in income positively correlate with cognitive scores and hippocampal volumes more in the lower (≤$75k) than higher income subsample (>$75k). While income is the primary measure of interest, we also report estimates and statistics from models fit using parental education in Supplementary Tables 2)[44–47].

To preview, we find that family income correlates with gains in memory and vocabulary scores, and the benefits of income on cognition are strongest in children whose families earn ≤$75k annually. We also observe that lower income is related to smaller volume of the stress-sensitive anterior (but not posterior) hippocampus. This relationship persists up until an annual income of ~$75k. Additionally, we find that the anterior hippocampus mediates income gaps in memory and vocabulary scores in children whose families earn ~≤$75k annually. Family income, therefore, selectively influences the volume of the stress-sensitive anterior but not posterior hippocampus, adding anatomical specificity to current theories. Our findings suggest that the anterior hippocampus may be a potential factor contributing to income differences in cognition.

## Results

**Income correlates positively with memory and vocabulary**. We first confirmed our a priori hypothesis that family income was positively related to episodic memory and vocabulary scores across the sample. Similar to previous work[1], we observed a positive relationship between family income and episodic memory scores, $b = 1.67$, SE = 0.31, $t(686) = 5.44$, $p < 0.001$, $p$-adjusted < 0.001, $r = 0.20$, and vocabulary scores, $b = 0.28$, $t(686) = 8.88$, $p < 0.001$, $p$-adjusted < 0.001, $r = 0.32$ (Fig. 1d, e for a visualization, and Supplementary Tables 3 and 4 for all model estimates). Of note, the $r$ value refers to the effect size of the relationship between the two measures described[48] (here, income and cognitive scores) after controlling for age and sex. We also observed a positive relationship between family income and processing speed scores ($b = 0.82$, SE = 0.30, $t(686) = 2.68$, $p = 0.008$, $p$-adjusted = 0.008, $r = 0.10$; Supplementary Fig. 12a) suggesting, as other studies have[1], that

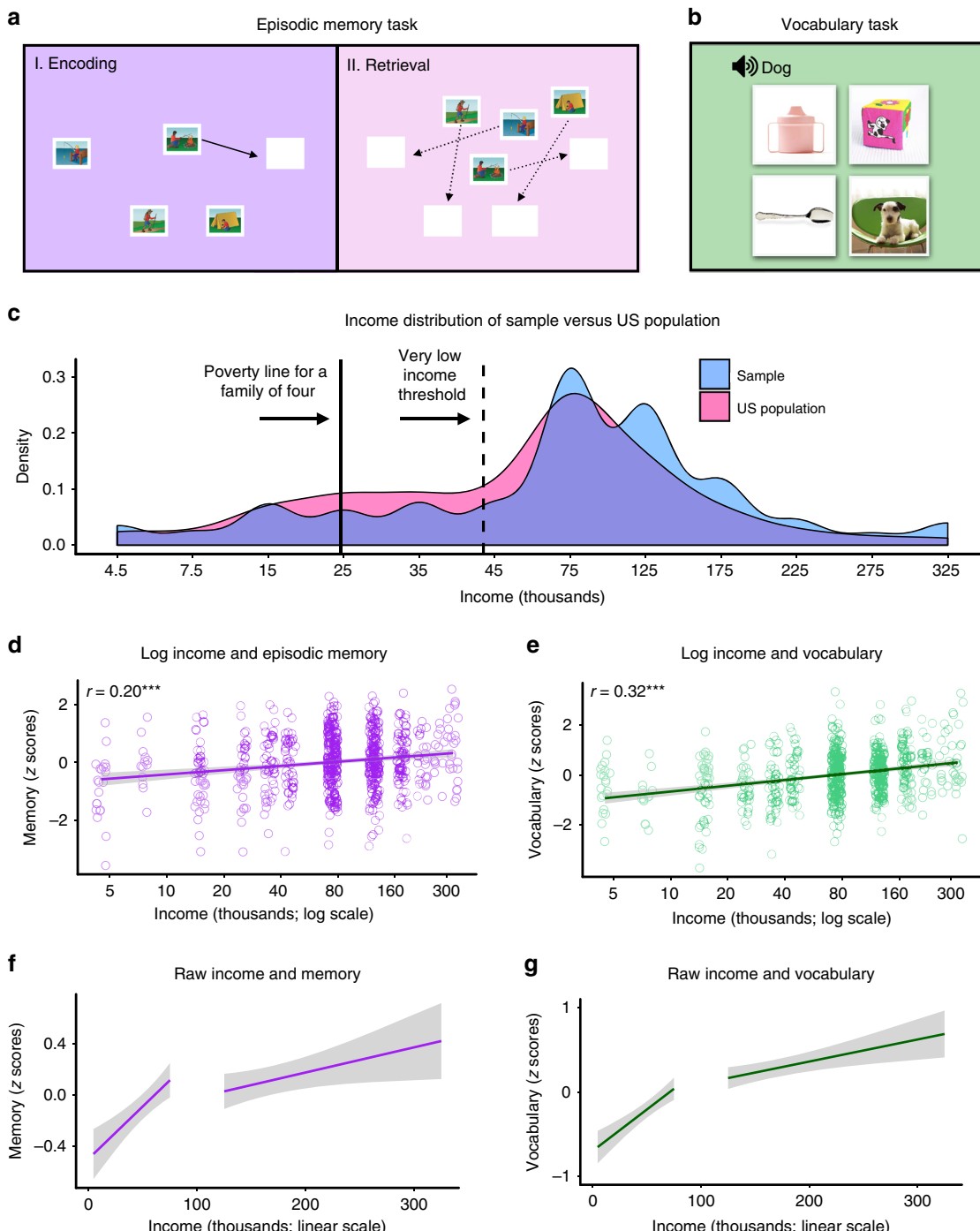

**a** Episodic memory task

I. Encoding   II. Retrieval

**b** Vocabulary task

🔊 Dog

**c** Income distribution of sample versus US population

Poverty line for a family of four

Very low income threshold

Sample
US population

Income (thousands)

**d** Log income and episodic memory

r = 0.20***

Income (thousands; log scale)

**e** Log income and vocabulary

r = 0.32***

Income (thousands; log scale)

**f** Raw income and memory

Income (thousands; linear scale)

**g** Raw income and vocabulary

Income (thousands; linear scale)

income has broad effects on cognition beyond cognitive processes supported by the hippocampus.

Using estimates from models that relate income to episodic memory and vocabulary scores, we determined that memory scores in the poorest children lag behind the wealthiest by approximately 4.5 years and vocabulary scores show a similar lag of approximately 5.5 years (Supplementary Fig. 5 and Supplementary Methods). Critically, we also found that age did not moderate the relationship between income and memory scores, $b = 0.05$, SE $= 0.06$, $t(685) = 0.81$, $p = 0.420$, $p$-adjusted $= 0.51$, $r = 0.03$, or income and vocabulary scores, $b = 0.006$, SE $= 0.006$, $t(685) = 0.93$, $p = 0.354$, $p$-adjusted $= 0.46$, $r = 0.04$, suggesting that income does not influence cognition differently across development (see Supplementary Tables 36, 37).

**Income predicts cognition more in poor than wealthy children.** We next asked whether income was more strongly related to cognitive scores in the lower (≤\$75k) than higher income (>\$75k) subsample. Consistent with this possibility, income gains corresponded to larger increases in cognitive scores among individuals in the lower than higher income subsample (interaction for memory: $b = -0.00002$, SE $= 0.000008$, $t(684) = -3.03$, $p = 0.003$, $p$-adjusted $= 0.007$; $r = 0.11$; interaction for vocabulary: $b = -0.000003$, SE $= 0.0000008$, $t(684) = -3.64$, $p < 0.001$, $p$-adjusted $= 0.002$; $r = 0.14$, Supplementary Tables 12–13, Fig. 1f, g). Notably, however, income gains correlated with better cognitive scores in both subsamples, albeit to a lesser degree in the higher income subsample (memory scores in lower income: $b = 2.25$, SE $= 0.48$, $t(395) = 4.66$, $p < 0.001$, $p$-adjusted $< 0.001$,

**Fig. 1 Task schematic, sample income distribution, and income−cognition relationships. a** During the Picture Sequence Memory Test, participants first encoded a series of thematically related images that appeared one at a time in the center of a computer screen (dark purple). After each image was displayed, it moved to a unique spatial location in the order in which it was presented. After all images were presented, the retrieval phase began (light purple). The images appeared in a scrambled position and participants were asked to move each image back to its original spatial location. This procedure repeated three times. **b** The Picture Vocabulary Test was used to assess vocabulary. Participants listened to an audio recording of a word on each trial while viewing four pictures. Participants were asked to select the picture that best matched the meaning of the word. **c** Income distribution of the sample (blue) versus the US population in 2012 (pink), retrieved from: https://www.census.gov/data/tables/time-series/demo/income-poverty/cps-finc-01.2012.html. Y-axis reflects percentages. Solid line reflects the United States poverty line for a family of four. Dashed line marks the average threshold for very low income status used for determining eligibility for assisted housing in 2012 within participants' metropolitan areas; retrieved from: https://www.huduser.gov/portal/datasets/il/il2012/select_Geography.odn. **d, e** Linear regressions showed income (log) was related to memory ($p < 0.001$) and vocabulary scores ($p < 0.001$), $n = 690$ participants. Income is plotted on a log scale reflecting our use of log transformed income. **f, g** The relationship between income and memory and vocabulary scores plotted on a linear scale, separately for lower and higher income subsamples. Linear regression interaction models showed income in raw dollars had a stronger relationship to cognitive scores in the lower ($\leq$\$75k) than higher income subsample (> \$75k; interaction for memory: $p = 0.003$, interaction for vocabulary: $p < 0.001$), $n = 690$ participants. For (**d**, **e**–**g**), the residuals of cognitive scores were calculated by removing variance related to age and sex and were transformed to z-scores for plotting. Gray shading reflects 95% confidence intervals around the mean. False discovery rate adjusted $p$ values are reported in the main text.

$r = 0.23$; vocabulary scores in lower income: $b = 0.25$, SE $= 0.05$, $t(395) = 4.85$, $p < 0.001$, $p$-adjusted $< 0.001$, $r = 0.24$; memory scores in higher income: $b = 2.85$, SE $= 1.40$, $t(287) = 2.04$, $p = 0.043$, $p$-adjusted $= 0.07$, $r = 0.12$; vocabulary scores in higher income: $b = 0.44$, SE $= 0.14$, $t(287) = 3.18$, $p = 0.002$, $p$-adjusted $= 0.005$, $r = 0.18$, Fig. 1f, g; Supplementary Tables 20–23). In contrast, there were no differences in relationships between processing speed scores in the lower versus higher income group (interaction: $b = -0.000003$, SE $= 0.000008$, $t(684) = -0.43$, $p = 0.667$, $p$-adjusted $= 0.71$, $r = 0.02$).

**Minority status does not moderate income−cognition relations**. Notably, the relationship between income and cognitive scores persisted after controlling for minority status ($ps < 0.001$) and were not moderated by the minority status ($ps$ of interaction with minority status $> 0.25$). This highlights the generalizability of the relationship between income and cognitive scores to minority and non-minority status individuals (see Supplementary Tables 48–53).

**Income positively predicts anterior hippocampal volumes**. To determine whether family income was also associated with the volume of the anterior and posterior hippocampus, we segmented the hippocampus into anterior and posterior subdivisions (see "Methods"). Consistent with our predictions, family income was associated with the volume of the anterior, $b = 43.20$, SE $= 12.17$, $t(693) = 3.55$, $p < 0.001$, $p$-adjusted $= <0.001$, $r = 0.13$, but not posterior hippocampus, $b = -14.88$, SE $= 10.48$, $t(693) = -1.42$, $p = 0.156$, $p$-adjusted $= 0.16$, $r = 0.05$ (see Fig. 2b and Supplementary Tables 5–6). We also observed an interaction between hippocampal subregion and family income, $b = -58.12$, SE $= 16.03$, $t(1394) = -3.63$, $p < 0.001$, $p$-adjusted $< 0.001$, $r = 0.10$, such that family income had a more positive influence on anterior than posterior volumes (Fig. 2b and Supplementary Table 7). Critically, age did not moderate the income−volume relationship for the anterior, $b = -1.52$, SE $= 2.35$, $t(692) = -0.64$, $p = 0.519$, $p$-adjusted $= 0.61$, $r = 0.02$, or posterior hippocampus, $b = -1.36$, SE $= 2.03$, $t(692) = -0.67$, $p = 0.501$, $p$-adjusted $= 0.60$, $r = 0.03$, suggesting that the effect of income on anterior and posterior hippocampal volumes was not different across development (see Supplementary Tables 38, 39 for model estimates and Supplementary Fig. 7 for a visualization). We also found that age was positively correlated with both anterior ($b = 11.31$, SE $= 2.58$, $t(693) = 4.38$, $p < 0.001$, $p$-adjusted $< 0.001$, $r = 0.16$) and posterior hippocampal volumes ($b = 18.54$, SE $= 2.22$, $t(693) = 8.34$, $p < 0.001$, $p$-adjusted $< 0.001$, $r = 0.30$; see Supplementary

Tables 5 and 6 for model estimates, and Supplemental Fig. 13a, b), but that age had a larger impact on posterior than anterior volumes ($b = 7.39$, SE $= 3.13$, $t(1394) = 2.36$, $p = 0.018$, $r = 0.06$; see Supplementary Table 60 for model estimates, and Supplementary Fig. 13c). Of note, minority status did not alter the relationship between age and anterior hippocampal volumes in the full or low-income subsample (see Supplementary Tables 58, 59).

**Income predicts volumes more in poor than wealthy children**. We next probed how income gains influenced hippocampal subregion volumes in the lower ($\leq$\$75k) as compared to higher (> \$75k) income subsamples. We found that the relationship between income and the anterior hippocampus was stronger in the lower income subsample, $b = 0.0008$, SE $= 0.0003$, $t(691) = 2.57$, $p = 0.01$, $p$-adjusted $= 0.07$, $r = 0.10$, Supplementary Table 14. Indeed, income gains only corresponded to increases in anterior hippocampal volumes in the lower, $b = 62.08$, SE $= 18.78$, $t(400) = 3.31$, $p = 0.001$, $p$-adjusted $= 0.01$, $r = 0.16$, but not the higher income subsample, $b = -4.12$, SE $= 58.21$, $t(283) = -0.07$, $p = 0.94$, $p$-adjusted $= 0.944$, $r = 0.004$, see Fig. 2c and Supplementary Tables 24, 25. Similarly, income gains had a greater impact on the posterior hippocampus among the lower as compared to the higher income subsample, $b = 0.0006$, SE $= 0.0003$, $t(691) = 2.06$, $p = 0.04$, $p$-adjusted $= 0.11$, $r = 0.08$, Fig. 2c and Supplementary Table 15. In contrast to findings in the anterior hippocampus, however, gains in income correlated with *decreases* in posterior hippocampal volumes in the lower income group, $b = -35.23$, SE $= 16.00$, $t(400) = -2.20$, $p = 0.028$, $p$-adjusted $= 0.09$, $r = 0.11$, Supplementary Table 26, with no relationship observed in the higher income group, $b = -4.42$, SE $= 51.12$, $t(283) = -0.09$, $p = 0.931$, $p$-adjusted $= 0.94$, $r = 0.005$, Fig. 2c, Supplementary Table 27. This bi-directional relationship highlights the importance of considering long-axis divisions in studies of how SES influences the development of the hippocampus. Notably, the relationship between income and hippocampal subregion volumes within the lower income group and across the full sample were not significantly moderated by minority status (lower income only: $ps > 0.20$, full sample: $ps > 0.07$), indicating no significant evidence that there are differences in income−volume relationships between minority status and non-minority status individuals. In addition, the significant effect of income on anterior hippocampal volumes is observed when controlling for minority status in the full and low-income sample, suggesting that these findings are present even after accounting for volume differences due to minority status (see Supplementary Tables 54–57 for estimates from models that include minority status and analyses split by minority status).

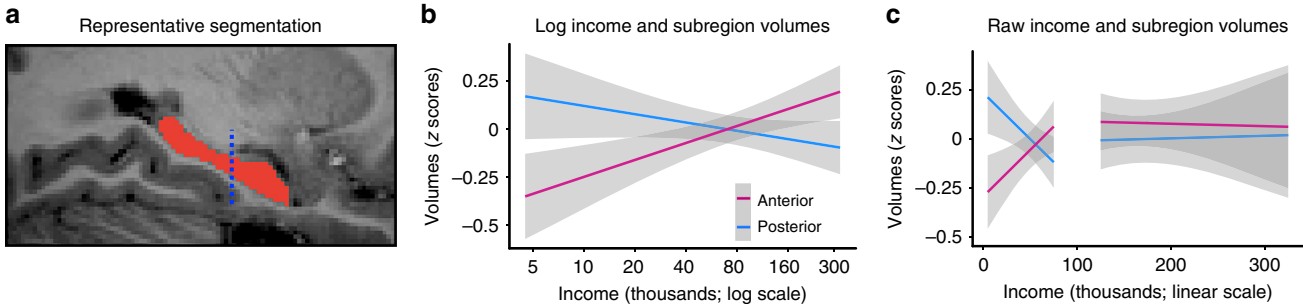

**Fig. 2 Hippocampal subregion volumes and their relationships to income. a** A segmentation of the hippocampus for a representative participant in the sagittal plane. An automated segmentation technique was used to segment the hippocampus and a manual segmentation approach was used to further subdivide the hippocampus into anterior and posterior segments at the uncal apex, marked by the dashed blue line. **b** Linear regressions revealed that income was related to the volumes of the anterior (pink), $p < 0.001$, but not posterior hippocampus (blue), $p = 0.16$. A linear mixed effects regression interaction model showed that income had a more positive influence on anterior than posterior hippocampal volumes, $p < 0.001$; $n = 703$ participants. See Supplementary Fig. 8a, b for plots with individual data points. We note that income is plotted on a log scale because a given income increment corresponded to larger gains in volume at lower ends of the income distribution. **c** The relationship between income and anterior (pink) and posterior hippocampal volumes (blue) plotted on a linear scale. Linear regression models showed that income in raw dollars had a stronger relationship to hippocampal subregion volumes in the lower (≤$75k) than higher income subsample (>$75k; interaction for anterior hippocampus: $p = 0.01$, interaction for posterior hippocampus: $p = 0.04$; $n = 703$ participants). For both plots in (**b**) and (**c**), we calculated residual values for volumes by removing the variance associated with age and sex. These residuals were then converted to $z$-scores for plotting. Gray shading reflects 95% confidence intervals around the mean. False discovery rate adjusted $p$ values are reported in the main text.

**Anterior hippocampal volumes positively predict cognition.** We next tested whether episodic memory and vocabulary scores were related to anterior and posterior hippocampal volumes. Although we did not have a priori predictions that episodic memory and vocabulary scores would be selectively related to a specific hippocampal subregion, consistent with prior work[23,49], we found that episodic memory and vocabulary scores positively associated with anterior (episodic memory scores: $b = 0.003$, SE $= 0.0009$, $t(686) = 2.79$, $p = 0.005$, $p$-adjusted $= 0.02$, $r = 0.11$; vocabulary scores: $b = 0.0004$, SE $= 0.0001$, $t(686) = 3.66$, $p < 0.001$, $p$-adjusted $= 0.002$, $r = 0.14$) but not posterior hippocampal volumes (episodic memory scores: $b = 0.0002$, SE $= 0.001$, $t(686) = 0.22$, $p = 0.828$, $p$-adjusted $= 0.83$, $r = 0.008$; vocabulary scores: $b = 0.00006$, SE $= 0.0001$, $t(686) = 0.52$, $p = 0.606$, $p$-adjusted $= 0.72$, $r = 0.02$, Fig. 3a, b, Supplementary Tables 8–11). As expected, neither anterior nor posterior hippocampal volumes correlated with processing speed scores (anterior: $b = 0.001$, SE $= 0.0009$, $t(686) = 1.51$, $p = 0.131$, $p$-adjusted $= 0.22$, $r = 0.06$, Supplementary Fig. 12b; posterior: $b = 0.002$, SE $= 0.001$, $t(686) = 1.45$, $p = 0.148$, $p$-adjusted $= 0.22$, $r = 0.06$), consistent with prior research that the hippocampus is involved in episodic memory[44,50] and vocabulary[41–43], but not processing speed[39,50].

Adding an age × volume interaction to models relating volumes to episodic memory and vocabulary scores revealed that anterior volume−cognition relationships did not depend on age (episodic memory scores: $b = -0.0002$, SE $= 0.0002$, $t(685) = -1.34$, $p = 0.179$, $p$-adjusted $= 0.47$, $r = 0.05$; vocabulary scores: $b = -0.000005$, SE $= 0.00002$, $t(684) = -0.27$, $p = 0.789$, $p$-adjusted $= 0.90$, $r = 0.01$; see Supplementary Tables 40, 41). In contrast, age moderated the relationship between posterior hippocampal volumes and cognitive scores, reflecting a more positive relationship between posterior hippocampus and cognitive scores among older than younger individuals (episodic memory scores: $b = -0.0007$, SE $= 0.0002$, $t(685) = -2.98$, $p = 0.003$, $p$-adjusted $= 0.04$, $r = 0.11$; vocabulary scores: $b = -0.00007$, SE $= 0.00002$, $t(685) = -2.85$, $p = 0.005$, $p$-adjusted $= 0.04$, $r = 0.11$, Supplementary Tables 42, 43). However, posterior hippocampal volumes were unrelated to cognitive scores within discrete age groups (young children: 3–7, older children: 8–12, adolescents: 13–17, young adults: 18–21), all $p$s > 0.10, see Supplementary Note 1.

Thus, while the direction of the relationship between posterior hippocampal volumes and cognitive scores may vary based on age, the size of the association is unreliable within discrete age groups. Of note, relationships between cognitive scores and volumes did not differ between higher and lower income subsamples, all $p$s > 0.27, Fig. 3c–f, Supplementary Tables 16–19 for interaction models and Supplementary Tables 28–35 for results examining volume−cognitive score relationships separately in the lower and higher income subsample.

**Anterior hippocampus mediates income-related cognitive gaps.** We then performed a mediation analysis to determine whether the shared variance between income and cognitive scores was explained by smaller anterior hippocampal volumes. We were particularly interested in the lower income individuals because the relationship between income and volumes and income and cognitive scores were strongest among lower income individuals. We observed that across the whole sample, anterior hippocampal volumes partially mediated income-related differences in episodic memory scores, $ab = 0.007$, SE $= 0.004$, 95% CI [0.0007, 0.020], and vocabulary scores, $ab = 0.007$, SE $= 0.004$, 95% CI [0.001, 0.020] (Fig. 4a, b). While the effects of family income on episodic memory scores, $c = 0.141$, SE $= 0.028$, 95% CI [0.085, 0.20], and vocabulary scores, $c = 0.191$, SE $= 0.025$, 95% CI [0.144, 0.24], were significant as expected, these relationships were reduced when taking anterior hippocampal volumes into account (episodic memory: $c' = 0.134$, SE $= 0.028$, 95% CI [0.079, 0.190]; vocabulary: $c' = 0.183$, SE $= 0.024$, 95% CI [0.137, 0.23]). Critically, this pattern of results replicated when only considering individuals in the lower income (Fig. 4c, d), but not higher income half of the sample. Indeed, within the lower income subsample, the positive relationship between episodic memory scores and income was mediated by the anterior hippocampus, $ab = 0.01$, SE $= 0.008$, 95% CI [0.0008, 0.03], as was the relationship between income and vocabulary scores, $ab = 0.01$, SE $= 0.006$, 95% CI [0.001, 0.03]. Though the relationship between income and memory and income and vocabulary was significant within the lower income subsample (memory: $c = 0.18$, SE $= 0.04$, 95% CI [0.09, 0.26], vocabulary: $c = 0.15$, SE $= 0.04$, 95% CI [0.08, 0.22]), these relationships were reduced after considering anterior hippocampus volume (memory: $c' = 16$, SE $= 0.04$, 95% CI [0.08, 0.25],

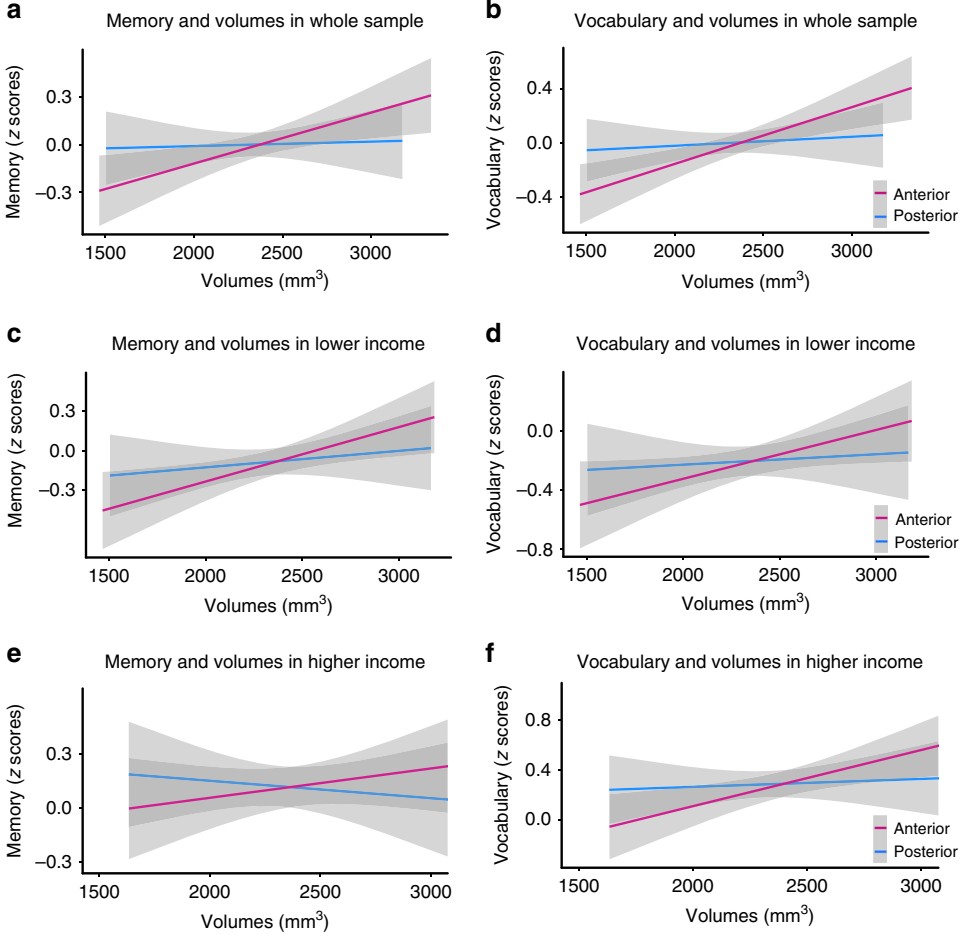

**Fig. 3 The relationship between cognitive scores and hippocampal subregion volumes. a**, **b** Linear regression models showed that across the sample, anterior (pink) but not posterior hippocampal volumes (blue) significantly correlated with episodic memory scores, $p = 0.005$, and vocabulary scores, $p < 0.001$, after controlling for age and sex; $n = 690$ participants. **c**, **d** Linear regressions also showed that within the lower income subsample (≤$75k), anterior (pink), but not posterior hippocampal volumes (blue) correlated with episodic memory, $p = 0.009$, and vocabulary scores, $p = 0.015$; $n = 399$ participants. **e**, **f** Within the higher income subsample, linear regressions showed that anterior hippocampal volumes (pink) correlated with vocabulary scores, $p = 0.01$, but not episodic memory scores, $p = 0.33$, $n = 291$ participants. There were no relationships between posterior hippocampal volumes (blue) and cognitive scores in either income subsample, all $ps > 0.40$. See Supplementary Fig. 9a–d for plots with individual data points. In all plots, the residuals of cognitive scores were calculated by removing the variance associated with age and sex. These residuals were then converted to $z$-scores for plotting. Gray shading reflects 95% confidence intervals around the mean. False discovery rate adjusted $p$ values are presented in the text.

vocabulary: $c' = 0.14$, SE = 0.04, 95% CI [0.07, 0.21]). In contrast, the anterior hippocampus in the higher income group (>$75k) did not mediate income-related gaps in memory scores, $ab = -0.001$, SE = 0.01, 95% CI [−0.02, 0.02], or vocabulary scores, $ab = -0.003$, SE = 0.02, 95% CI [−0.04, 0.02]. Although a cross-sectional sample constrains our ability to infer for causal relationships, these findings are consistent with the possibility that income differences in the anterior hippocampus partially account for worse memory and vocabulary scores among lower as compared to higher income individuals. Of note, neither posterior nor whole hippocampal volumes mediated the relationship between income and cognitive scores (see Supplementary Figs. 1–4), suggesting that the hippocampus' relationship with income-related cognitive gaps has been obscured by not taking these anatomical divisions into account.

## Discussion
Our results are consistent with the prominent theory that the hippocampus mediates income-related gaps in children's cognitive abilities[3,4,6,18]. By showing that anterior, but not posterior, hippocampus mediates income-related differences in cognitive

scores, this work adds a crucial anatomical distinction to these theories. Indeed, we found this division is so crucial that income had opposing relationships with anterior and posterior volumes in children from lower income homes. Of note, the relationship between income and hippocampal subregion volumes and income and cognitive scores did not depend on age, suggesting that the effects of income on cognition and the hippocampus are pervasive and consistent across development. We also observed that income's relationship with cognitive scores and anterior hippocampal volumes were strongest among individuals from a lower income background (i.e., children from families that earned ≤$75k annually). Crucially, this suggests that incremental gains in income benefit brain and cognitive development, up to a certain threshold (in our sample, ≤$75k annually), but may have diminishing benefits thereafter.

The large public dataset used here did not include a stress measure, so we were unable to directly assess the role that income-related stress played in our findings. Thus, while stress is a plausible mechanism, it is not the only factor that may influence the relations we observed. Indeed, many other factors, including access to material and nonmaterial resources[3,7] correlate with SES and may

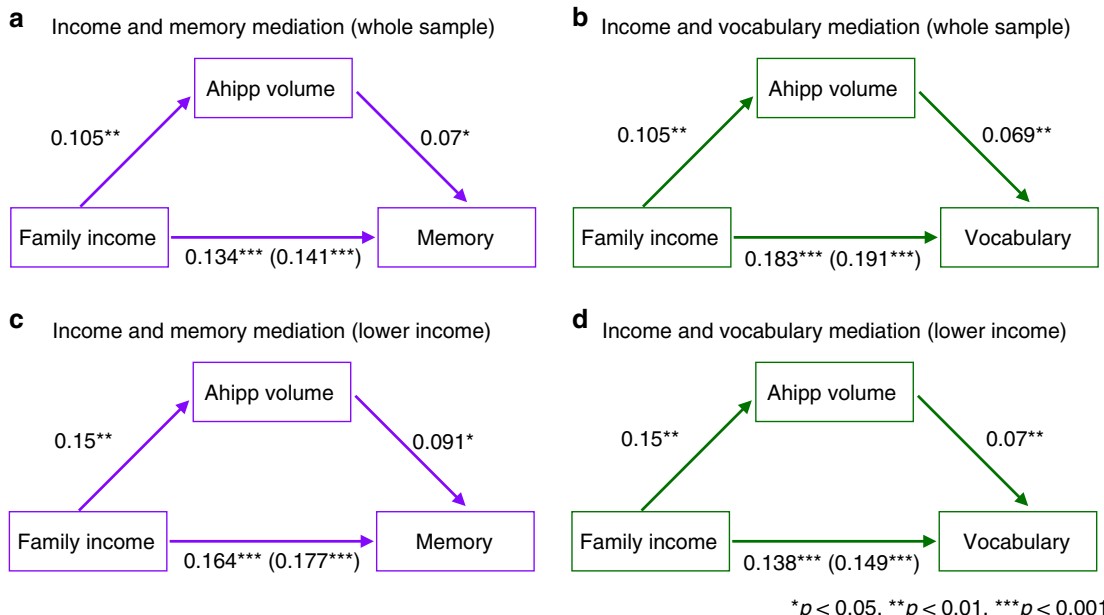

**Fig. 4 Depiction of anterior hippocampal mediation of income-cognitive score relationships. a**, **b** Bootstrapped mediation analyses revealed that across the sample, anterior (Ahipp) hippocampal volumes partially mediated income-related gaps in episodic memory, $p = 0.03$, and vocabulary scores, $p = 0.006$, $n = 690$ participants. **c**, **d** Similarly, within the lower income subsample, anterior (Ahipp) hippocampal volumes partially mediated income-related gaps in episodic memory, $p = 0.03$, and vocabulary scores, $p = 0.02$, $n = 399$ participants. Standardized beta values are reported. The values in parentheses are the standardized beta coefficients reflecting the relationship between the two variables *before* accounting for anterior hippocampal volumes (i.e., the total effect). The values in front of the parentheses are the relationship between the variables after accounting for anterior hippocampal volumes (i.e., the direct effect).

play a role in our findings. For example, relative to families from lower SES backgrounds, higher SES families spend more time engaging children in reading activities[51] and provide more access to educational resources[52], which are thought to be important for cognitive and neural development[53]. These factors could boost anterior hippocampal volumes and cognitive performance among individuals from higher income backgrounds, and thus underlie the positive effects of income on cognition and anterior hippocampal volumes that we observed. It is our hope that our correlative findings will inspire future longitudinal research to directly measure stress to determine whether stress or other factors related to income can prospectively predict changes in anterior hippocampal volumes during development. Moreover, although not available in this dataset, future work investigating the effects of income should consider using an income-to-needs ratio (i.e., income divided by the national poverty threshold for a family of the same size), which would provide a more precise estimate of the amount of resources available to children than income alone.

An additional critical question which our study is not powered to address is how minority status interacts with SES to produce effects on brain and cognitive development. Given prior research showing that minority groups from low SES backgrounds experience greater stress than non-minority status individuals from both high and low SES backgrounds in the United States[54], the negative effects of low SES on brain and cognition could be exacerbated among minority groups. Because our sample had few individuals from minority status backgrounds in the higher income range, we were not positioned to test these questions. However, our results showing that minority status did not moderate relationships between income and cognitive scores or income and volumes within the lower income subsample (≤$75k) suggest that relationships with income are not different between minority status and non-minority status individuals. Moreover, the fact that income remained a reliable correlate of anterior hippocampal volumes and memory and vocabulary after

controlling for minority status highlights the generalizability of our findings. We hope that these exploratory analyses motivate future studies to address questions about how minority status interacts with SES more thoroughly and to include more diverse samples in studies of brain development and cognition.

To conclude, our findings are relevant to clinicians, educators, and policy makers, who are interested in promoting brain and cognitive health in children from lower income backgrounds. Moreover, our results add the much-needed anatomical specificity to current theories by showing that family income disproportionately affects the anterior hippocampus. Crucially, the present study also highlights that the anterior hippocampus may be a potential mediator of cognitive gaps between high- and low-income children. Given that the anterior and posterior hippocampus are involved in different cognitive processes[27], the differential influence of income on the development of these hippocampal regions may have implications for understanding the types of cognitive processes that require more support in low-income children.

## Methods

**Participants**. To investigate the relationship between family income and hippocampal subregion volumes, we used data collected from the Pediatric Imaging, Neurocognition and Genetics study[55] (data now stored in the NIMH-supported Research Domain Criteria Database (RDoCdb)). The RDoCdb is a collaborative informatics system created by the National Institute of Mental Health to store and share data resulting from grants funded through the Research Domain Criteria project. Individuals were excluded from participating if they had been diagnosed with a developmental, psychiatric or neurological disorder, had a history of head trauma, were born premature, or had been exposed to drugs or alcohol prenatally for more than one trimester. The experimental conditions and consenting procedures were approved by the human research protection of research subjects and institutional review board at each participating site—namely, University of California San Diego, the University of Hawaii, University of California Los Angeles, University of California Davis, Kennedy Krieger Institute at Johns Hopkins, Sackler Institute at Cornell University, the University of Massachusetts, Massachusetts General Hospital at Harvard University, and Yale University. Participants and parents gave written informed consent/assent to participate in the study procedures, including cognitive

### Table 1 Sample demographics (*n* = 703).

|  | Mean (SD; Range) or *n* (%) |
| --- | --- |
| Age | 12.3 (5; 3–21) |
| Sex |  |
| Female | 338 (48%) |
| Male | 365 (52%) |
| Family income | $99,950 ($74,880); $4500−$325,000) |
| Genetic ancestry |  |
| African American | 74 (10.5%) |
| American Indian | 6 (<1%) |
| Asian | 59 (8%) |
| Hispanic | 33 (4.5%) |
| Pacific Islander | 2 (<1%) |
| White | 521 (74%) |

Note: Genetic ancestry data were missing for eight participants.

assessments, demographic questionnaires and structural MR scanning. In the present study, we only included participants who consented to share their raw neuroimaging data, and whose parents/guardians disclosed their annual family income (*n* = 750). Among these 750 participants, 47 were excluded for one of the following reasons: excessive motion/artifacts on MR scan (38 participants), poor segmentation of the hippocampus (2 participants), outlier (either anterior or posterior hippocampal subregion volumes were more than three standard deviations from the sample mean, seven participants). Therefore, 703 participants were included in the analyses investigating the relationship between family income and hippocampal volumes (see Table 1 for demographics). Among these 703 participants who had usable neuroimaging data, 690 completed cognitive assessments. Analyses on measures of cognitive performance included these 690 participants.

**Family income**. Parents were asked to report annual family income. In the case that parents did not accompany the participant to the lab, participants were asked to provide their parents combined income. Data were collected in bins that ranged from <$5,000 to over $300,000 per year (see Supplementary Table 1). Each income bin was re-coded as the average value of each bin. This value was then log transformed to remove the positive left skew in the data and to reflect the larger impact that small income gains might have for those in low as compared to high-income subsamples. Although an income-to-needs ratio would be an ideal measure of available resources, we did not have data on the number of individuals in each household and thus measures of income were used as our primary measure. Critically, family income was unrelated to age, $p = 0.885$, $r = 0.005$, and sex, $p = 0.296$, $r = 0.04$.

**Cognitive assessments**. Previous research suggests that the hippocampus is critical for tasks that depend on the ability to bind relational associations into memory[40]. The hippocampus is also thought to contribute to novel word learning[20–24], by associating novel words with a meaning or by extracting the meaning of a novel word from a broader semantic context[20–23]. Given the hippocampus supports these aspects of cognition that are vulnerable in lower income individuals[1,3], we analyzed participant data from the NIH toolbox's Picture Sequence Memory[56] and Picture Vocabulary Tests[57], which measure episodic memory and vocabulary abilities respectively (Fig. 1a, b).

The Picture Sequence Memory Test requires that participants associate an image with a particular temporal and spatial order and therefore is thought to rely on hippocampal binding[58]. The task is divided into an encoding and retrieval phase. During the encoding phase, participants viewed a sequence of thematically related images that appeared one at a time, in the center of the computer screen (2.2 s each). As each image appeared, an audio recording described the content of the image. After each image was presented, it was moved to a unique spatial position on the computer screen that matched the temporal order in which the images were presented. Thus, the spatial location of each image was correlated with the order in which the images were presented. The encoding phase ended once all of the images had been presented and moved to their unique spatial position. To adjust for age-related increases in episodic memory ability, image sequence length varied from 6 to 15 pictures, depending on the age of the participant. The retrieval phase began 3 s after the encoding phase ended. During retrieval, all images appeared on the computer screen in a scrambled order, and participants were asked to move each image back to its correct spatial position. Participants were allowed as much time as needed to complete the retrieval phase. Participants encoded and retrieved the sequence of images three times, with improved performance on each repetition dependent on long-term memory. Episodic memory scores consisted of the total number of pairs of images that were correctly placed adjacent to each other during the retrieval phase. Although this task correlates with standardized measures of episodic memory[56,58], we note that there is only a short delay between encoding and retrieval, and consequently, working memory processes may

contribute to task performance. Critically, however, the hippocampus is required for binding temporal and spatial relationships—even over short delays[40]—making it an excellent task for assessing hippocampal function. Because this task targets short-term memory for associations, the generalization of these findings to other aspects of memory—for example, long-term autobiographical memory—remains to be tested. However, validation studies of this task have shown that this task has sufficient construct validity and correlates well with performance on other measures of memory[58]. Although the hippocampus is a key region for supporting memory for temporal and spatial relationships[40], it is likely not the only brain region that contributes to task performance. Indeed, both the inferior frontal gyrus and prefrontal cortex[59,60] have been shown to be important for episodic memory during development.

During the Picture Vocabulary Task[57], participants listened to an audio recording of a word on each trial, while viewing four images on a computer screen. Participants were asked to select the image on the computer screen that best matched the meaning of the spoken word. We included this task because low SES status is associated with poor language functioning[1]. Although language is represented by many brain regions, including the prefrontal and temporal cortices[61], the hippocampus is believed to be particularly involved in the acquisition and use of novel vocabulary[20–24], making it a useful task for understanding how income differences in the hippocampus influence cognition. This task was administered with computer adaptive testing, which allows for the difficulty of the words presented to be tailored to the ability of the participant. Test difficulty is adapted such that participants have a 50% chance of answering correctly on each trial.

In order to determine whether hippocampal volumes and income broadly correlate with cognition, or selectively correlate with cognitive abilities known to be supported by the hippocampus (e.g., memory and vocabulary), we also included a processing speed task[47] which is thought to be independent of the hippocampus[39,50].

On each trial of the processing speed task[47], participants were presented with two images in the center of the computer screen and had to decide whether the images were the same, or not the same, by pressing a button to indicate yes (the images are the same) or no (the images are not the same). The images were simple line drawings that depicted common things (e.g., clouds, trees), and were either identical, or differed on one of three dimensions: (1) color; (2) adding/taking something away; (3) one versus many. Scores on the task reflected the number of correct items (of a possible 104 for ages 3–7 and 130 for ages 8–15) that participants completed in 90 s.

**Image acquisition and processing**. For each of the 750 participants who met the inclusion criteria (i.e., had measures of family income and consented to providing raw imaging data), we accessed their raw structural MR data that was acquired using 3 T MRI scans (either Siemens, Philips, or General Electric) across the participating sites. All protocols included a sagittal 3D T1-weighted whole-brain inversion prepared recovery spoiled gradient echo scan for gray matter segmentation, and prospective motion correction[62]. Data were acquired in the sagittal plane with interleaved slice acquisition. Identical, or nearly identical, protocols were used across sites and scanners to reduce the effects of scanner on imaging measures (for the Siemens scanners: TE = 4.33 ms, TR = 2170 ms, flip angle = 7°, voxel size = 1 × 1 × 1.2 mm voxels, FoV = 256; matrix size = 256 × 256; for the Philips scanner: TE = 3.1 ms, TR = 6.8 ms, flip angle = 8°, voxel size = 1 × 1 × 1.2 mm voxels, FoV = 256; matrix size = 256 × 240; for the General Electric scanner: TE = 3.5 ms, TR = 8.1 ms, flip angle = 8°, voxel size = 1 × 1 × 1.2 mm voxels, FoV = 256, matrix size = 256 × 192). However, we observed that scanner had a significant influence on measures of hippocampal volume, and thus, scanner was included as a covariate in all analyses in which volume was a dependent variable.

Prior to segmenting the hippocampus, we visually inspected the images to check for visual features of motion, including motion rings and ghosting outside of each MR image using Display (version 2.0), an MRI image viewing software (Information on Display viewer and download instructions are available online: http://www.bic.mni.mcgill.ca/software/Display/Display.html). Images were categorized based on the degree to which features of motion were detectable on a scale of 1–4, that included no signs of motion (1), minimal signs of motion (2), clear signs of motion (3) or excessive motion (4). Images that had either clear or excessive signs of motion were excluded, whereas images with no signs of motion or minimal motion were included in analyses. In total, 38 participants were excluded at this stage for clear and excessive signs of motion, leaving 712 eligible participants.

**Hippocampal segmentations**. We used a combination of automatic and manual methods to define the anterior and posterior hippocampus. First, we segmented the whole hippocampus automatically, using the Multiple-Automatically Generated Templates for different brains algorithm (MAGeT Brain)[63]. This approach has been validated in clinical and healthy adult samples and generates labels for the whole hippocampus that are comparable to existing automated methods[63]. The MAGeT Brain algorithm uses a set of manually labeled hippocampal atlases as inputs to segment unlabeled T1 images in a dataset. We used five pre-existing, manually segmented hippocampal atlases (Winterburn atlases) that included definitions of hippocampal subfields as inputs[64], which have previously been used in analyses validating MAGeT Brain[63], and which span the length of the anterior

−posterior hippocampal axis. The MAGeT Brain algorithm registers these manually labeled atlas labels via nonlinear image registration to a subset ($n = 21$) of MR images in the sample specified as template images. Each of the newly generated labels on each template image is then registered to the entire dataset of MR images. The labels on each MR image are then fused using a voxel voting procedure, in which the most commonly occurring label at each voxel is retained as part of the final label. By registering the atlases to a subset of the sample, the templates, and then using the template labels to segment the entire dataset, labeling errors that might arise due to anatomical differences between the atlases and subject images are minimized[63]. Although we are unaware of a study that has validated these atlases in a developmental sample, hippocampal atlases developed by the same group that are based on adult anatomy and used in conjunction with MAGeT[63] have been validated in a developmental sample[65]. This study showed that MAGeT in conjunction with adult hippocampal atlases produce accurate labels in a developmental sample relative to manually derived labels[65]. This suggests that using adult atlases to segment the developing hippocampus with the MAGeT algorithm results in labels that have acceptable accuracy relative to manually derived labels.

Once automated segmentation of the whole hippocampus was complete, we combined the subfield labels (CA1, CA2-3, CA4-DG, subiculum, SRSLSM) into a single label for the left and right hippocampus. We combined the subfield labels because whole hippocampal volumes using this method are more reliable than those generated for the individual subfields[63], and because we were particularly interested in examining the anterior and posterior hippocampus segments (rather than the subfields). After labels were generated, we visually inspected each to ensure that the label covered the hippocampus. Data were included if the segmentations covered the entire hippocampus on each slice that the hippocampus was visible, or the majority of the hippocampus on each slice that the hippocampus was visible. Otherwise, data were excluded. In the case of two labels, the segmentations only covered a few (3 or 4) slices of hippocampus. These two labels were excluded, leaving a total of 710 labels. To ensure objectivity for excluding these two labels, we selected a random subset of 100 labels among the 712 MR images (including the two labels that we decided to exclude due to poor segmentation). We then had a second blind rater successfully identify these two poor quality labels (and only these labels) for exclusion.

To extract measures of the anterior and posterior hippocampus, a trained rater (A.L.D.) identified the slice that subdivided the anterior and posterior segments on the 710 remaining images. This was done by identifying the slice corresponding to the uncal apex, which is a commonly used landmark for the anterior−posterior hippocampus boundary[66]; see Fig. 2a for a visualization that marks the uncal apex. The caudal-most slice of the anterior hippocampus corresponded to the last slice at which the uncal apex was visible, and the rostral most slice of the posterior hippocampus corresponded to the first slice at which the uncal apex was no longer visible. To ensure the accuracy of this boundary, a second trained rater re-identified the anterior−posterior boundary in an overlapping 10% of the labels. For the anterior−posterior boundary, the researchers identified the same slice in 94% of cases, and the same or a slice that differed by one slice in 98% of cases. Both raters were blind to demographic information relevant to this study (age, sex, family income). After identifying the boundary slice, any part of a subfield label (CA1, CA2/3, CA4/DG, subiculum, SRLM) that fell rostral to the uncus was counted towards the volume of the anterior hippocampus, whereas any part of a label that fell caudal to the uncus was counted towards the volume of the posterior hippocampus. Thus, the volume of the anterior and posterior segments, respectively, reflected the total voxels covered by any subfield label that was rostral or caudal to the boundary slice. In this way, the subfield labels were largely treated as though they were a single label: we ignored the divisions between them, and they were not used to inform the boundary between anterior and posterior segments.

To ensure that our findings were unbiased by participants' head size, we adjusted the volume of all regions for individual differences in intracranial volume. We used measures of ICV that were already acquired by the Pediatric Imaging, Neurocognition, and Genetics study using a specialized processing stream[55]. To correct regional volumes for intracranial volume, we used a regression approach[67]. In this analysis, we regressed intracranial volumes onto regional hippocampal volumes (left and right anterior, and posterior), such that the residual value (the regions size minus its predicted value based on each individual's intracranial volume) was accounted for in each region for each individual (see Supplementary Methods). Before correcting for ICV, we fit four separate linear regression models to test whether age or sex interacted with ICV to predict left and right anterior and posterior hippocampal volumes. While sex did not interact with age or ICV to predict volumes, the interaction between ICV and age was significant for left and right anterior hippocampal volumes. Therefore, we divided our sample into subsamples based on age for the anterior hippocampus and performed the ICV correction separately on these subsamples. In adjusting the volume of the right and left anterior hippocampus, younger and older children (3–7, 8–12 years of age) were combined into a single group because the relationship between ICV and volumes did not differ in younger and older children. For the same reason, adolescents and young adults (13–17, 18–21 years of age) were combined into a different group for ICV correction. Since neither age nor sex interacted with ICV to predict posterior hippocampal volumes, we did not divide the sample into subsamples to adjust posterior hippocampal volumes for ICV (see Supplementary Methods).

After correcting for ICV, we tested whether there were age and hemisphere-related effects on volumes before combining volumes across hemispheres. In one set of models, we tested whether hemisphere interacted with age to predict volumes, and in another set of models, whether income interacted with hemisphere to predict volumes (i.e., age × hemisphere; income × hemisphere interactions). We did this separately for the anterior and the posterior hippocampus. We found that the relationship between age and hippocampal subregion volumes did not differ by hemisphere (all $ps > 0.58$). This suggests that while hippocampal subregion volumes get larger as children get older, this effect does not differ significantly by hemisphere. Similarly, we found that hemisphere did not moderate the relationship between income and hippocampal subregion volumes (all $ps > 0.76$), suggesting that the relationship between income and volumes did not differ across the two hemispheres. Since there were neither age × hemisphere nor income × hemisphere interactions, we calculated bilateral hippocampal volumes by summing analogous regions in the left and right hemispheres. Bilateral hippocampal volumes that had been adjusted for ICV were used in all analyses. Of note, seven participants whose anterior or posterior hippocampal volumes fell three standard deviations from the sample mean were excluded leaving a total of 703 participants in the final analyses.

**Statistical analyses**. All statistical analyses were carried out in R (version 3.6.3)[68]. Data are available through the National Institute of Mental Health (NIMH) Data Archive (NDA), Dataset identifier(s): [10.15154/1519020][69]. For all statistical tests, age was mean centered, sex was effect-coded (female = −1, male = 1), scanner was dummy coded, and income was log transformed. All analyses were adjusted for the family wise false positive discovery rate using false discovery rate (FDR) correction. We report adjusted $p$ values that have been corrected for multiple comparisons in the text after each uncorrected $p$ value. Of note, exploratory analyses were corrected for multiple comparisons separately from confirmatory tests. $p$ values of <0.05 were considered statistically significant. Two-tailed tests were used for all analyses.

**Family income−cognition relationships**. We first performed analyses to test whether family income correlated with children's and adolescents' episodic memory, vocabulary and processing speed scores. Analyses on income-processing speed relationships were used to determine whether income−cognition relationships were restricted to hippocampal-mediated cognitive processes (e.g., memory and vocabulary) or related to broader aspects of cognition thought to be hippocampal independent. Thus, we ran three separate linear regressions, in which family income was used as a predictor, and episodic memory, vocabulary, and processing speed scores from the full sample were entered as dependent variables in separate models. All models included age and sex as covariates. For completeness, we also report results from analogous models that include nonlinear age transformations as covariates in the Supplementary Note 2.

We also added additional exploratory analyses. First, we added an age × income interaction to the income−cognition models to test whether the relationship between income and cognition varied as a function of age. We also examined whether relationships between income and cognition were stronger in a lower (≤$75k) versus higher (>$75k) income subsample by adding an income subsample × income interaction term to the above models. To explore potential interactions, we also examined income−cognition relationships separately in the lower (≤$75k) and higher income (>$75k) subsample. Last, we examined whether minority status (i.e., white vs. non-white) moderated relationships between income and memory and vocabulary measures by adding a minority status × income interaction to these models. To explore the effects of minority status further, we also re-ran models assessing income−memory and income−vocabulary relationships for minority status and non-minority status individuals separately. Since there were few individuals from minority status backgrounds in the higher income subsample (>$75k; $n = 40$ minority status), we restricted these analyses to individuals who made $75k and less annually ($n = 134$ minority status, $n = 268$ non-minority status).

**Family income−hippocampal subregion volume relationships**. We next evaluated whether family income was associated with anterior or posterior hippocampal volumes. We ran two separate linear regressions that included data from the full sample, in which anterior and posterior hippocampal volumes served as dependent variables in separate models. In both models, we included the log transformation of family income, age, sex, and scanner number as predictors/covariates. Prior to entering age into the model, we assessed whether the linear, quadratic or cubic age term best fit the data. We used the most parsimonious age term, unless a more complex term fit the data better at $p < 0.05$. Neither the quadratic nor cubic terms fit the data better than the linear effect of age, and therefore we retained the linear age term in both models. In addition, we tested whether family income influenced anterior and posterior hippocampal volumes differently. Therefore, we constructed a linear mixed effects model, in which volumes served as the dependent variable, and region (dummy coded: anterior = 0, posterior = 1) and the log transform of family income served as predictors and interaction terms. Because regions were nested within participants, we

modeled participants using a random intercept to account for the random effect of participant on regional volumes (see Supplementary Table 7).

In addition to the above models, we explored whether age moderated income–volume relationships by adding an age × income interaction to the above models. We also examined whether income–volume relationships were more prominent in the lower than higher income subsamples (≤$75k vs. >$75k) by adding an income subsample × income interaction term our models. To probe differences in higher and lower income groups further, we examined income–volume relationships in the lower and higher income subset separately. We also examined how minority status interacted with income to influence hippocampal subregion volumes by adding a minority status × income interaction to the above models. To explore potential effects of minority status further, we also re-ran these models separately for non-minority status and minority status individuals in the lower income subsample.

Finally, since patterns of age-related differences in hippocampal subregions are not well established, we assessed whether age influenced hippocampal subregion volumes more in the posterior than anterior hippocampus. We ran an interaction mixed effects model to test whether age had a more positive relationship on one subregion than another. For the interaction mixed effects model, since regions were nested within participants, we modeled random intercepts for each participant.

**Volume–cognition relationships**. We further examined whether anterior and posterior hippocampal volumes were associated with episodic memory and vocabulary. We also tested whether hippocampal subregion volumes correlated with measures of processing speed to determine whether hippocampal subregion volume–cognition relationships were selective to aspects of cognition known to be hippocampal-dependent. We ran six separate linear regression models that included data from the full sample. Anterior and posterior hippocampal volumes were used as predictors in separate models, and episodic memory, vocabulary and processing speed scores were used as dependent variables in separate models. Of note, we controlled for the linear effect of age in these models because we wanted to test whether the hippocampus correlated with cognitive scores above and beyond experience (age). Whereas the linear age term is a proxy for experience overtime (which is inherently linear), quadratic and cubic nonlinear transformations of age may well be driven by variance in the brain across individuals. Given we had a direct measure of the brain, we did not add nonlinear age transformations as covariates to these models. For completeness, we report results from analyses that include nonlinear age transformations as covariates in the supplement (see Supplementary Note 2).

In addition, we explored whether age moderated relationships between volume and memory and vocabulary by adding an age × volume interactions to the above models. Furthermore, we tested whether relationships between volume and memory and vocabulary differed in the lower vs. higher income subsamples. To further explore the effects of minority status, we also re-ran models examining relationships between volume and episodic memory and vocabulary scores separately for minority status and non-minority status individuals in the lower income subsample. All models included age and sex as covariates.

**Do anterior hippocampal volumes mediate gaps in cognition?** Finally, we were interested in whether hippocampal volumes mediated income-related gaps in cognition. Given the relationships between income and hippocampal volume were restricted to the anterior hippocampus in the full sample, we were primarily interested in whether anterior hippocampus mediated income-differences in cognition. Therefore, we performed two separate mediation analyses to examine the direct and indirect effects of income on episodic memory and vocabulary scores. The direct and indirect effects of income and memory and vocabulary scores were modeled using linear regressions[70]. We controlled for age, sex, and scanner in all models. For each mediation analysis, we ran 5000 bootstrap samples to provide stable estimates of the direct, indirect and total effects. We report 95% confidence intervals and considered intervals that did not include zero statistically significant. We also explored whether posterior or whole hippocampal volumes mediated income-gaps in cognition. These analyses were performed in the same way as the mediation analyses for the anterior hippocampus. We report the results from these exploratory models in the supplement, Supplementary Figs. 1–4. Furthermore, we also explored whether anterior hippocampus mediated income-related cognitive gaps in the lower income subsample (≤$75k; Fig. 4c, d).

**Reporting summary**. Further information on research design is available in the Nature Research Reporting Summary linked to this article.

## Data availability
The data that support the findings of this study are available from the NIMH-supported Research Domain Criteria Database. Dataset identifier(s): [https://doi.org/10.15154/1519020]. Restrictions apply to the availability of these data, which were used under license for the current study, and so are not publicly available. Data are however available from the authors upon reasonable request and with permission of NIMH. A reporting summary for this Article is available as a Supplementary Information file.

## Code availability
The code that was used to analyze the data are publicly accessible at the following link: https://github.com/alexandradecker/PING_script.

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

## Acknowledgements

We would like to thank Sarah Berger and Hannah Cho for their work on proofreading this manuscript. Also, we are grateful to Jovanka Skocic, Jiin Kim and Carissa deMarinis for visually inspecting the hippocampal labels and MR images and Bharat Nadendla for his help compiling income data for the population of the United States. Data and/or research tools used in the preparation of this manuscript were obtained and analyzed from the controlled access datasets distributed from the NIMH-supported Research Domain Criteria Database (RDoCdb). RDoCdb is a collaborative informatics system created by the National Institute of Mental Health to store and share data resulting from grants funded through the Research Domain Criteria (RDoC) project. Dataset identifier (s): [https://doi.org/10.15154/1519020]. This manuscript reflects the views of the authors and may not reflect the opinions or views of the NIH or of the Submitters submitting original data to RDoCdb. This work was supported by Brain Canada.

## Author contributions

A.L.D., K.D., A.S.F. and D.J.M. conceived the idea. All authors contributed equally to the hypotheses. A.L.D. performed the analyses and wrote the manuscript. All authors contributed equally to editing the manuscript.

## Competing interests

The authors have no competing interests.
