## [Peer Review File · Nature Communications]

Editorial Note: Parts of this Peer Review File have been redacted as indicated to remove third-party material where no permission to publish could be obtained

Reviewers' comments:

Reviewer #1 (Remarks to the Author):

This is study of hippocampal volume, SES, and memory/language in a large group of participants. My enthusiasm for this work, elegant analyses, and large sample size is tempered by the fact that many assumptions that are made are not tested. There are a lot of correlational findings.

1. The authors write, "These cognitive gaps coincide with differences in brain
30 structure, likely driven by the effects of income-related chronic stress on brain
31 development^{2,5,17}. One brain region – the hippocampus – has been singled out by
32 prominent theories because it is highly sensitive to chronic stress^{5–8}, is consistently
33 smaller in children with low SES^{2,3,5,11,17}, and is important for both memory⁹ and
34 language performance¹⁰." and in the Discussion, "This work raises the possibility that chronic stress
in
162 childhood degrades anterior hippocampal structure, triggering a cycle of altered stress
163 responses, and further structural degradation of the hippocampus and other brain
164 regions."

This seems to be the crux of this paper. However, stress is NOT measured, with either verbal report questionnaires or hair/blood/nail cortisol. There is no evidence that income and stress levels correlate in this sample. Indeed, they don't in many, if not most samples that range this widely in income (5-345K, Mean = 99K). This is a major and faulty assumption. If not stress, then why look at the hippocampus?

2. In fact, the sample is so high in SES (income is close to 100,000\$) that to interpret the effects as having anything to do with low income samples is inaccurate. Figure 1 d and e are not compelling. There is a lot less data, making it susceptible to small individual differences, in the lower income ranges. The distribution is clearly skewed toward higher income children. No normalizing will fix the interpretation. Perhaps a more accurate interpretation is that anterior hippocampal volume in a high SES sample is related to memory and language. This is not about poor vs. wealthy children. The effects in Fig 1 and 2 are linear and the sample is high SES on average.

3. Age range is so wide 3-21 and so little is said about this. This is important for the reasons stated next.

4. The relationship between the tasks used and hippocampal volume is not cleanly established. Episodic memory tasks and language broadly involve hippocampus + IFG, PFC (Often et al, 2007; Sowell et al., 2001), and other aspects of the temporal lobes. This is not minor. Functional MRI data. with these tasks

is needed to establish the assumption that the tasks link to the anterior hippocampus uniquely or primarily. Especially for the language measure. Indeed the existence of that relationship (language as a mediator of SES and hippocampal volume) calls into question the episodic memory result and suggests the anterior hippocampal findings may be correlates of other regions possibly connected to anterior but not posterior hippocampus.

5. Indeed, evidence for the relationship between hippocampal volume and episodic memory is mixed and variable across the age range tested. Recent data from Demaster & Ghetti (2013) showed in adults, but not in children, better episodic memory performance was associated with smaller right hippocampal head and larger hippocampal body. In children, but not in adults, better episodic memory was associated with larger left hippocampal tail. How does this work with the large age range tested here? And the anterior hippocampal result?

In sum, the authors may be completely right. However, there are sufficient untested claims here that may be correlational third-variable confounds. Given the importance to society of getting these findings right, more work is needed.

Reviewer #2 (Remarks to the Author):

This is a review of the manuscript titled “Income-related gaps in children’s cognition mediated by anterior not posterior hippocampus” by Decker et al. The authors examine possible relationships between income, cognition, and structural measures of the hippocampus (including anterior and posterior hippocampus), in a large sample of participants from a wide age range (ages 3-21 years). The large sample size is a noted strength as it could allow testing income-related environmental effects on brain, on cognition, and on interesting interactions. Yet, the wide age range does not play as a strength given it is largely ignored. Assessing age-related effects on income, brain and behavior interactions could strengthen the manuscript as would clarifying the theoretical framing. Below I list concerns regarding the lack of clear theoretical framing and methodology.

Concerns about lack of theoretical framing

The choices of measures used here were not sufficiently motivated. For example, I find the evidence justifying a focus on anterior vs. posterior hippocampus to be fairly weak, obtained with cross-sectional samples. Such lack of theoretical framing, in turn, limits robust interpretation. For example, when the authors note (pages 5-6): “Last, we asked why the anterior hippocampus is correlated with both income and cognition. These correlations may reflect independent mechanisms, a common cause of income, or – of most theoretical interest – the anterior hippocampus’ role in mediating the relationship between income and cognition.” The authors neglect to list many other possibilities that they could have tested to assess the significance of the significant mediation effect they later describe. This mediation effects (especially in cross sectional design) falls far short from establishing an answer to a ‘why’ question. Overall, I strongly advocate for the authors to carefully amend the text to remove statements such as this (page 8): “These findings support theoretical models of income-related cognitive gaps by directly

demonstrating the role of the hippocampus in shaping cognitive differences between wealthy and poor children.” There is nothing in the data that lend support for such strong statement. Specifically, the findings here do not demonstrate causality as is implied by the use of the word ‘shaping’. At the very least toning down is strongly advised.

Methods/Results

1. Lacking clear theoretical framing and given that developmental differences in hippocampal structure are not well established, the authors should test (establish) age differences in hippocampal anterior/posterior measures in this large sample. The authors include age as a covariate in all analyses. However, age-related effects should be of interest on its own (see comment about rationale above). Previous research has identified different sensitivity of the hippocampus to SES over age in developing sample. With a big sample of wide age range, the authors are privileged to examine age-related effect on hippocampal subregions and cognition, differential SES effect on hippocampal subregions at different ages, e.g., potential age x SES interactive effect on the hippocampus. Moreover, these age-related effects may differ along the long-axis of the hippocampus, if the posterior hippocampus shows more protracted development. These analyses would provide more comprehensive picture of SES effect on hippocampal subregions across age and are crucial for drawing adequate interpretations.

2. Stating limitation in the tasks used:

a. Language task seems to be merely about vocabulary. More broadly, it is unclear why language measures are included in an investigation of hippocampal SES relations. Furthermore, the authors interpretation of the significant relation between language ability and anterior hippocampus is not clear.

b. Memory task is quite a short-term / working memory rather than an episodic memory task. Moreover, the authors don’t provide a justification for the limitation of this specific task in representing ‘memory’ more broadly. This task has spatial recall component. What if this task characteristic, rather than a more general ‘memory’ process makes it selectively related to anterior Hc and not posterior hippocampus?

3. Missing information: On page 3, the authors mentioned “These model estimates demonstrate that memory ability in the poorest children appears to lag behind the wealthiest children by 4.5 years and language ability shows a similar lag of 5.5 years”. Please explain how the numbers of years were calculated. Regardless, authors should note the limitation of this interpretation based on cross-sectional study to the authors.

4. On page 9, the authors mention “Notably, protective factors, such as cognitive stimulation, greater material resources, and better education, could boost hippocampal volumes in wealthier children, and thus play a role in the income-volume relationships that we observed. However, this possibility is inconsistent with our results, which showed that income-volume relationships were steepest at the lower end of the income distribution.” As the relation between income and volume is linear (no significant quadratic effect), how is this ‘steepest’ conclusion drawn?

5. As for the measure of income:

a. First, why income? The authors decision to use income level as the only proxy of SES is not sufficiently motivated. Other measures, such as parental education (for enriched environment and parent-child interaction), social class or even composite score should also be considered.

- b. If income, then income-to-need ratio should be used instead, given that available resources to depends on the number of people in the household.
- c. Are the SES of adults (18-21 yr) their own or their parents? If the latter, did the parents visit the lab? Please clarify the procedure.
6. More information should be provided on the selection of approach used for hippocampal segmentation. For example, were the 5 atlases used validated in developing sample and used T1 images as used here? Related, the authors should better clarify anterior and posterior segmentation boundaries.
7. When conducting regressions for ICV correction across the whole sample, the assumption of homogeneity of regression should be checked. Given this is a sample with wide age range, the relation between ICV and ROI volumes may differ depending on age; if this is the case, the ICV correction should be conducted within sub-samples of different age. This may also be true for sex.
8. When conducting separate linear regression models, multiple comparison correction should be conducted to avoid increase rate of false positive, e.g., page 16.

Minor points:

1. For figure 2 b,c,d, presenting individual data in scatterplots will greatly help the reader get a sense of the distribution and appreciate the findings depicted in the figures.
2. The table of demographic can be moved to the main text. At least, information about age (mean, SD, range) should be mentioned at "participants" section.

Reviewer #3 (Remarks to the Author):

The authors find that the anterior hippocampus size, and not the posterior hippocampus, is partially mediating a relationship between income variation and memory and vocabulary abilities in children. While the relation between SES and cognitive achievement is well-established, this manuscript is an important advancement regarding the mixed findings in the hippocampus. The use of a well-powered dataset combined with an analysis that follows many best practices for data transparency and analysis is to be commended. The manuscript is extremely well written, reasonable, and clear; it is a compelling work that will have substantial impact and be of use to multiple research areas in development and cognition. I have a few points for consideration, clarification or enhancement.

1. Income is but one proxy for socioecological context. Often SES impact can be an accumulation of different risk factors, of which income is but one. Was race ever examined as a factor or controlled for in the analysis? While this study appears to use a majority white sample, perhaps at least minority status could be considered as a moderating factor of interest on the income-hippocampal-cognition relation. Were any other SES factors such as parental education, single parenting, use of food stamps/SNAP, family mobility, zip code, or housing collected? Relatedly, as these data were collected from different scanners, and different parts of the country, quality of life can vary substantially for the same income (money can go much farther in Kansas than San Diego). How was this addressed in your sample? Some

comments as to the limitations of using a single measure, such as income, as a proxy for SES should be added to the manuscript if other factors are not available to include in the model.

2. Greater clarity is needed as to how “good parcellation” was determined, and “low movement” was determined from the scans. What was the “excessive” motion that resulted in removal, and how did it differ from acceptable motion? Was a rating system or metric used for either type of QA (motion or parcellation)? If so, how were different quality levels determined? If not, how was consistency determined across scans, and how poor of quality was still acceptable? Similarly, on page 13, “a visual inspection was performed to ensure the quality of all images in the analysis”. The presence of what factors required removal from analysis, more specifically?

3. Minor points: Line 111 (page 6) refers to the wrong figure. Line 156 (page 8) is missing a “this” in “Consistent with possibility”.

Reviewers' comments:

Reviewer #1

This is study of hippocampal volume, SES, and memory/language in a large group of participants. My enthusiasm for this work, elegant analyses, and large sample size is tempered by the fact that many assumptions that are made are not tested. There are a lot of correlational findings.

1. The authors write, "These cognitive gaps coincide with differences in brain 30 structure, likely driven by the effects of income-related chronic stress on brain 31 development^{2,5,17}. One brain region – the hippocampus – has been singled out by 32 prominent theories because it is highly sensitive to chronic stress^{5–8}, is consistently 33 smaller in children with low SES^{2,3,5,11,17}, and is important for both memory⁹ and 34 language performance¹⁰." and in the Discussion, "This work raises the possibility that chronic stress in 162 childhood degrades anterior hippocampal structure, triggering a cycle of altered stress 163 responses, and further structural degradation of the hippocampus and other brain 164 regions."

This seems to be the crux of this paper. However, stress is NOT measured, with either verbal report questionnaires or hair/blood/nail cortisol. There is no evidence that income and stress levels correlate in this sample. Indeed, they don't in many, if not most samples that range this widely in income (5-345K, Mean = 99K). This is a major and faulty assumption. If not stress, then why look at the hippocampus?

The reviewer raises an important point that the relation between stress and income is less clear in samples with broad income ranges such as ours. Indeed, stress does *not* correlate with income among high-income earners only (Kahneman & Deaton, 2010; Jebb, Diener & Oishi, 2018, which we now discuss on page 4, lines 76-84 of the manuscript). Kahneman & Deaton (2010) found that income correlated with everyday stressful experiences up to an annual income of ~\$75k, and Jebb, Diener & Oishi (2018) similarly found that higher income correlated with fewer stressful experiences, up to an annual income of ~ \$95k. These findings are consistent with the reviewer's point that income-stress relationships don't always hold in broad income samples. We thank the reviewer for highlighting this important consideration and now include new analyses to take this factor into account.

In particular, we divided our sample into lower (\leq \$75k) and higher ($>$ \$75k) income groups and tested whether income-cognition and income-volume relationships differed in the two subsamples. If the reported effects are influenced by stress, then relationships with income should be more pronounced in the lower than higher income group. We found that this was the case: income-

volume and income-cognition relationships were stronger in the lower income group. Examining these income groups separately showed that in the lower income group, family income positively correlated with anterior hippocampal volumes, and cognition, and the anterior hippocampus mediated income-related gaps in cognition. While income positively correlated with cognition in the higher income sample, it was unrelated to anterior hippocampal volumes, which did not mediate income-cognition relationships. These findings are consistent with the stress mechanism that motivated our analyses, and also have important implications for understanding the distinct mechanisms that may underlie income-cognition relationships in the different groups. For these reasons, we have now reframed the paper around this important finding, discussing these differences on page 6, lines 107-122; pages 8-9, lines 174-192; page 10, lines 204-211, as well as the abstract, lines 18-20. We also report statistics and estimates from these models in the supplement (pages 54-61, Supplementary Tables 12-19; 20-35).

We also take the reviewer's concern about inferring a stress mechanism very seriously. As outlined in our original manuscript, our project was motivated through the lens of a stress mechanism: children living in lower-income homes are likely to experience more stress (as measured by questionnaires and hormones) than their higher income peers (Felner, Brand, DuBois, Adan, Mulhall & Evans, 1995; McLoyd, 1998; Evans & English, 2002; Bradley & Corwyn, 2002; Evans & Kim, 2007; Evans & Cassels, 2013; Luby, Belden, Botteron, Marrus, Harm, Babb, Nishino & Barch, 2013) and the anterior divisions of the hippocampus are more sensitive to stress than the posterior divisions (Szeszko et al., 2006; Vythilingam et al., 2005; O'Leary & Cryan, 2014; Tanti & Belzung, 2013; Tanti et al., 2012; Willard et al., 2009; Hawley et al., 2012). This literature inspired our investigation of anterior-posterior hippocampal subdivisions and resulted in the important discoveries reported in our manuscript. Given the substantial literature showing that individuals from higher income backgrounds have less stress than those from lower income backgrounds (Evans & English, 2004; Kim, Evans, Angstadt, Shaun Ho, Sripada, Swain, Liberzon & Phan, 2013; Evans & Cassels, 2013; Luby et al., 2013; Kim, Capistrano & Congleton, 2016; Algren, Ekholm, Nielsen, Ersboll, Bak & Anderson, 2018), we respectfully maintain that it is theoretically important to situate our research within this context. We do, however, agree with the reviewer that it is equally important to alert the reader to the fact that we do not directly measure stress and that without this measurement, stress remains strictly a potential, plausible mechanism. We state this directly on page 15, lines 291-293: "*The large public dataset used here did not include a stress measure, limiting our ability to directly assess the role of income-related stress, per se. Thus, stress remains a plausible mechanism. However, our findings are consistent with literature that inspired our work that directly measures or manipulates stress.*" And on page 15, lines 300-303: "*It is our hope that our correlative findings will inspire future longitudinal research that directly measures stress responses in children to determine whether stress can prospectively predict changes in anterior hippocampal volumes.*"

Further, we carefully considered each reference to stress in our manuscript, clarifying when it refers to the prior literature or is an inferred plausible mechanism. When the latter, we now couch statements in terms of plausible inferences, making it clear that mechanisms cannot be directly supported in this dataset.

2. In fact, the sample is so high in SES (income is close to 100,000\$) that to interpret the effects as having anything to do with low income samples is inaccurate. Figure 1 d and e are not compelling. There is a lot less data, making it susceptible to small individual differences, in the lower income ranges. The distribution is clearly skewed toward higher income children. No normalizing will fix the interpretation. Perhaps a more accurate interpretation is that anterior hippocampal volume in a high SES sample is related to memory and language. This is not about poor vs. wealthy children. The effects in Fig 1 and 2 are linear and the sample is high SES on average.

We also would like to thank the reviewer for raising these thoughtful points. Upon re-reading the manuscript, we now see how we led the reader to believe that income was linearly related to behaviour and the brain, and further that this effect could be driven by children in the higher income group. Specifically, we failed to emphasize that none of our analyses were conducted using raw income (in dollars) – we used the log-transformation of family income. We performed this transformation to capture the intuitive idea (borne out in our data) that the addition of a given increment of income to a lower-income household will have a greater impact than adding that same increment to a high-income household. This log transformation is consistent with the methodology used in prior studies that examine how income influences measures of well-being and stress (Kahnemann & Deaton, 2010; Jebb, Diener & Oishi, 2018) as well as brain and cognitive health (Noble et al., 2012). We have now added a figure in the manuscript to show readers that relationships between income and cognition and income and subregion volumes are steeper in individuals from families earning \$75k/year or less than those earning >\$75k/year (Figure 1f & g; Figure 2c; see figures pasted at end of this response), as well as a figure in the supplement to visualize the relationship in discrete income subsamples (Supplementary Figure 13, page 70; see figure at end of this response). Thanks to the reviewer, rather than just reporting this log transformation, we now emphasize it both in the figure legends and text, (page 4, lines 79-87; page 8, lines 149-154; page 10, lines 201-204) as well as provide a description of its implications for the interpretation of the full sample results (page 4, lines 79-84): *“Of note, the log transformation of family income was used in all analyses to reflect the idea, borne out in our data, that a given income increment to a lower income family has more impact than adding the same increment to a higher income family. Thus, in our analyses, the linear relationships reported reflect the greater impact that income increments have on lower as compared to higher income individuals.”*

More importantly, this and the above comment motivated us to perform analyses that directly assess how lower- and higher-income families contribute to the reported effects. As described above, we now report the significant interaction between income group ($\leq \$75k$ vs. $> \$75k$) and income, which demonstrates income-cognition and income-hippocampal volume relationships are, indeed, more pronounced among children from lower income families (See Figure 1f & g and 2c pasted below). We also replicated all analyses in the lower-income group ($\leq \$75k$). We report the results of these analyses on page 6, lines 107-122; pages 9-10, lines 174-192; page 10, lines 206-208, page 11, line 236). The statistics and model estimates are also detailed in Supplementary Tables 12-19 and 20-35).

Although there are fewer individuals in the lowest income bracket ($< \$4.5$) the overall sample of individuals whose household income is equal to or less than $\$75k$ is large ($n= 410$) and shows reliable income-related increases in cognition and anterior hippocampal volumes. Notably 83 of these children come from homes with incomes under the US poverty line for a family of 4. Moreover, based on municipal income thresholds that were used for determining eligibility for assisted housing in 2012, 169 participants are considered *very low* or *extremely low*-income in their respective metropolitan areas (data retrieved from the 2012 HUD Metropolitan Fair Market Rent Income Limits Database; link: https://www.huduser.gov/portal/datasets/il/il2012/select_Geography.odn). Thus, interpretations relevant to children living in poverty or low-income environments are unlikely to be driven by a few outlying data points. To improve readers' ability to discern sample characteristics pertaining to income, we have now added a table to the supplement detailing the number of individuals in each income bin and the cumulative sum (See Supplementary Table 1, page 46). We have also removed any direct mention of the term poverty from the manuscript so as to not mislead readers as to the nature of this sample. Thank you for highlighting this very important point.

Figure 1 f & g) The relationship between household income and memory and vocabulary plotted on a linear scale. Income in raw dollars had a stronger relationship to cognitive scores in the lower ($\leq \$75k$) than higher income subsample ($> \$75k$; interaction for memory: $p = .003$, interaction for vocabulary: $p < .001$). The residual values of cognition scores were calculated by removing variance related to age, and sex, and then transforming the residuals to z-scores.

Figure 2c) The relationship between family income and anterior and posterior hippocampal volumes plotted on a linear scale. Income in raw dollars had a stronger relationship to hippocampal subregion volumes in the lower ($\leq \$75k$) than higher income subsample ($> \$75k$; interaction for anterior hippocampus: $p = 0.01$, interaction for posterior hippocampus: $p = 0.035$).

Supplementary Figure 13. Plotted is income in raw dollars against A) memory and B) language scores to illustrate the larger impact that income gains had on the lower than higher half of the income distribution. To reflect the idea, borne out in our data, that incremental gains in income influenced cognition more in the lower ($\leq \$75k$) than higher ($> \$75k$) half of the sample, the log transformation of income was used in all analyses.

3. The relationship between the tasks used and hippocampal volume is not cleanly established. Episodic memory tasks and language broadly involve hippocampus + IFG, PFC (Often et al, 2007; Sowell et al., 2001), and other aspects of the temporal lobes. This is not minor. Functional MRI data with these tasks is needed to establish the assumption that the tasks link to the anterior hippocampus uniquely or primarily. Especially for the language measure. Indeed, the existence of that relationship (language as a mediator of SES and hippocampal volume) calls into question the episodic memory result and suggests the anterior hippocampal findings may be

correlates of other regions possibly connected to anterior but not posterior hippocampus.

We thank the reviewer for raising this point about the tasks used in the study. We would first like to address the reviewer's concern that we are assuming that the episodic memory and vocabulary tasks link primarily or uniquely to the anterior hippocampus. To clarify, we are not assuming that these tasks are solely supported by the anterior hippocampus and completely agree that other brain regions likely influence task performance. To clarify this point for readers, we have pointed out in the manuscript that we are not purporting that these tasks link primarily to the anterior hippocampus and reference work mentioned by the reviewer showing that other brain regions are involved on page 20, lines 404-407: *"Moreover, although the hippocampus is a key region for supporting memory for temporal and spatial relationships⁵³⁻⁵⁶, it is likely not the only brain region that contributes to task performance. Indeed, both the inferior frontal gyrus and prefrontal cortex^{72,73} have been shown to be important for episodic memory during development."* We also note that other brain regions support language on page 20, lines 404-407, *"Although language is represented by many brain regions, including the prefrontal and temporal cortices⁷⁴, the hippocampus is believed to be particularly involved in the acquisition and use of novel vocabulary²⁴⁻²⁸, making it a useful task for understanding how income differences in the hippocampus influence cognition."*

The second point that we would like to address is that it is necessary to show that the tasks uniquely or primarily depend on the anterior hippocampus. We do not agree that this is a necessary prerequisite to answering the question: does the anterior rather than posterior hippocampus mediate SES related gaps in cognition? Rather, we selected tasks which should involve the hippocampus, potentially both anterior or posterior divisions, and which show robust relationships with SES. Combined, these two requirements are sufficient to determine whether income has a greater impact on anterior as compared to posterior divisions, and whether this strong impact mediates the influence income holds on task performance.

We would also like to emphasize that there is a large body of research implicating the hippocampus in tasks like the two included here. The episodic memory task requires participants to bind temporal and spatial associations into memory. Relational binding – even over short delays – is thought to depend on an intact hippocampus (Cohen, Poldrack & Eichenbaum, 1997; Konkel & Cohen, 2009; Olson et al., 2006; for a review see Olsen, Moses, Riggs, & Ryan, 2012; Moscovitch, Cabeza, Wincour & Nadel, 2016), and studies examining hippocampal subregions show that relational binding is supported by the anterior hippocampus (Sperlin, Cocchiarella, Rand-Giovannetti, Poldrack, Schacter & Albert, 2003; Giovanello, Schnyer & Verfaellie, 2003; Jackson & Schacter, 2004; Chua, Schacter, Rand-Giovannetti & Sperling, 2007; Staresina & Davachi, 2009).

Moreover, the task was designed specifically so that the number items to be recalled exceed the limits of children and adults' working memory capacity, so that the task loads onto episodic rather than working memory (Dikmen et al., 2014). We have highlighted that the relational binding component of this task in the manuscript, page 18, lines 368-371: "*Previous research suggests that the hippocampus is critical for tasks that depend on the ability to bind relational associations into memory...*"; and page 19, lines 377-379 "*The Picture Sequence Memory Test requires that participants associate an image with a particular temporal and spatial order and therefore is thought to rely on hippocampal binding*".

There is also evidence that the hippocampus supports performance on the type of low-frequency vocabulary task used here. In the vocabulary task, participants hear a word, see four images and are asked to select the image that best matches the spoken word. Task difficulty is controlled by adjusting word frequency so that participants have a 50% percent chance of answering correctly on each trial. Even though the hippocampus is not classically studied in relation to language, aspects of language, like learning low frequency word meanings, are thought to depend on hippocampal binding (James & MacKay, 2001; Davis & Gaskell, 2009; Shtyrov, 2011; Duff & Brown-Schmidt, 2012). In particular, hippocampal binding is thought to facilitate new word learning by linking a novel word to a meaning or by extracting the meaning of a novel word from a context of known words (James & MacKay, 2001; Davis & Gaskell, 2009; Duff & Brown-Schmidt, 2012). Studies linking the hippocampus to word learning show that hippocampal activity correlates with both novel word learning and semantic knowledge (Breitenstein, Jansen, Foerster, Wolbers & Knecht, 2005; Davis, Di Betta, Macdonald, Gaskell, 2009) and that vocabulary training increases hippocampal volume (Bellander, Berggren, Martensson, Brehmer, Wenger, Li, Bodammer, Shing, Werkle-Bergner & Lovden, 2016). Most importantly, individuals with hippocampal dysfunction have difficulty with learning new words (Gabrieli, Cohen & Corkin, 1988; James & MacKay, 2001; Warren & Duff, 2014). To clarify our rationale for using the language task for readers, we have referenced additional papers in the manuscript that highlight the role that the hippocampus plays in vocabulary learning, page 2, line 38, "*is important for both episodic memory²³ and aspects of language²⁴⁻³⁰*", page 4, lines 69-71: "*We chose to include episodic associative memory and vocabulary assessments in our investigation because both memory⁵²⁻⁵⁶ and the acquisition and use of new vocabulary^{24,25,27,28,57-61} are thought to depend on hippocampal binding*", and page 18, line 369-371: "*The hippocampus is also thought to contribute to novel word learning²⁶⁻³⁰, by associating novel words with a meaning or by extracting the meaning of a novel word from a broader semantic context²⁶⁻²⁹.*"

4. Age range is so wide 3-21 and so little is said about this. This is important for the reasons stated next.

5. Indeed, evidence for the relationship between hippocampal volume and episodic memory is mixed and variable across the age range tested. Recent data from Demaster & Ghetti (2013) showed in adults, but not in children, better episodic memory performance was associated with smaller right hippocampal head and larger hippocampal body. In children, but not in adults, better episodic memory was associated with larger left hippocampal tail. How does this work with the large age range tested here? And the anterior hippocampal result?

We thank the reviewer for asking whether volume-cognition relations may differ across development. In response, we tested whether age moderated our reported effects and we now report the results of these important analyses in the manuscript on pages 5-6 (lines 101-106), 9 (lines 167-173), 11 (lines 220-236) and provide full details in the Supplementary Materials (Page 62-66, Supplementary Tables 36-44, Supplementary Figures 7 & 8). To summarize, we found no evidence that age moderated anterior volume-cognition relationships (our primary effect of interest). Interestingly, however, age did moderate posterior-volume cognition relationships, such that the relationship between posterior hippocampal volume and cognition was more positive in older than younger children. Examining the relationship between posterior volumes and memory in age groups separately (young children: 3-7, older children: 8-12, adolescents: 13-17; young adults: 18-21) revealed that cognition-volume relationships were not statistically significant in any group. Thus, posterior volumes do not seem to reliably correlate with memory across different age groups. Although our results differ from Demaster & Ghetti (2013), our sample size is well powered to detect reliable effects, and is much larger than their sample, which comprised of 23 adults and 34 children after exclusions. We have cited their work to acknowledge our differing results (page 45, lines 973-975) “*We note that our results might differ from previous studies examining how age moderates volume-cognition relationships⁸³, however, owing to the large sample size in our study, we are well powered to detect significant effects*”.

In sum, the authors may be completely right. However, there are sufficient untested claims here that may be correlational third-variable confounds. Given the importance to society of getting these findings right, more work is needed.

We have read your comments and suggestions with great interest and have revised our paper accordingly. In response to your concern that income does not correlate with stress in samples such as ours that vary so widely in income, we have tempered our conclusions relating to stress in the manuscript, and have also demonstrated that income-volume and income-cognition relationships not only hold within a subsample of individuals from lower-income backgrounds, but are stronger within this as compared to the higher income group. Your comments have also motivated us to demonstrate that the reported results hold across different age groups and we have detailed the results of these analyses in the text for readers. We also better justify our choice of cognitive measures. Overall,

we feel the paper is much stronger and complete because of your comments and we greatly appreciate your time and thoughtfulness.

Reviewer #2

This is a review of the manuscript titled “Income-related gaps in children’s cognition mediated by anterior not posterior hippocampus” by Decker et al. The authors examine possible relationships between income, cognition, and structural measures of the hippocampus (including anterior and posterior hippocampus), in a large sample of participants from a wide age range (ages 3-21 years). The large sample size is a noted strength as it could allow testing income-related environmental effects on brain, on cognition, and on interesting interactions. Yet, the wide age range does not play as a strength given it is largely ignored. Assessing age-related effects on income, brain and behavior interactions could strengthen the manuscript as would clarifying the theoretical framing. Below I list concerns regarding the lack of clear theoretical framing and methodology.

Concerns about lack of theoretical framing

The choices of measures used here were not sufficiently motivated. For example, I find the evidence justifying a focus on anterior vs. posterior hippocampus to be fairly weak, obtained with cross-sectional samples. Such lack of theoretical framing, in turn, limits robust interpretation. For example, when the authors note (pages 5-6): “Last, we asked why the anterior hippocampus is correlated with both income and cognition. These correlations may reflect independent mechanisms, a common cause of income, or – of most theoretical interest – the anterior hippocampus’ role in mediating the relationship between income and cognition.” The authors neglect to list many other possibilities that they could have tested to assess the significance of the significant mediation effect they later describe. This mediation effects (especially in cross sectional design) falls far short from establishing an answer to a ‘why’ question. Overall, I strongly advocate for the authors to carefully amend the text to remove statements such as this (page 8): “These findings support theoretical models of income-related cognitive gaps by directly demonstrating the role of the hippocampus in shaping cognitive differences between wealthy and poor children.” There is nothing in the data that lend support for such strong statement. Specifically, the findings here do not demonstrate causality as is implied by the use of the word ‘shaping’. At the very least toning down is strongly advised.

Your concern about a lack of theoretical framing is very helpful to us. From the beginning, we were interested in investigating the role of anterior and posterior regions of the hippocampus in mediating the relationship between income and cognition because of the extensive animal literature indicating that the hippocampus—especially the anterior hippocampus—is sensitive to stress. We now cite additional studies motivating our focus on anterior and posterior hippocampus

specifically. Because of the limits on the number of references in our initial submission (only 20 allowed for main text) we originally omitted these papers. We now reference papers that directly manipulate stress to assess how stress is *causally* related to changes within the anterior hippocampus on page 3, lines 49-54, (Willard et al., 2009; Perera, et al., 2011; O’Leary, O’Connor, & Cryan, 2012; Hawley & Leasure, 2012; Tanti et al., 2012; Hawley, et al., 2012; Tanti & Belzung et al., 2013; O’Leary & Cryan, 2014). We have also cited papers from human work, including longitudinal data from individuals with Cushing syndrome (a disease characterized by over-secretion of glucocorticoid stress hormones; Starkman et al., 2014), major depression (associated with hypersecretion of glucocorticoids), and post-traumatic stress disorder (Szeszko et al., 2006; Vythilingam et al., 2005) on page 3, lines 53-60. We think that the inclusion of these additional papers, help to motivate our focus on the anterior hippocampus and enhance the theoretical framing the study. We also add additional text on page 3, lines 49-58 to motivate our study:

“Indeed, work in both humans and animals show that stress preferentially impacts the anterior hippocampus^{16,17,34-39}. Stressed animals display disproportionate decreases in cell survival and neurogenesis in the ventral (anterior) portion of the hippocampus^{34-36,38-40}, and in humans^{16,17} and nonhuman primates³⁷, stress is linked to smaller anterior hippocampal volumes. Furthermore, antidepressants – a common treatment for stress related disorders – selectively increases neurogenesis in the anterior hippocampus^{34-36,40}. Likewise, longitudinal work in humans shows that resolving pathological over-secretions of stress hormones selectively increases anterior, but not posterior hippocampal volumes¹⁸. The ventral (anterior) hippocampus is also involved in regulating stress and anxiety related behaviors^{41,42}, and physiological responses to stress^{43,44} via dense structural connectivity with the amygdala⁴⁵, and hypothalamic endocrine and autonomic nuclei^{33,46}.”

We have also now removed statements that imply causality, such as “shaping cognitive differences” and replaced them with tempered language, noting the limitations of the cross-sectional sample. On page 13, lines 262-265, we state: *“Although a cross-sectional sample constrains our ability to infer for causal relationships, these findings are consistent with the possibility that income differences in the anterior hippocampus partially account for worse memory and vocabulary abilities among lower as compared to higher income individuals”.*

Methods/Results

1. Lacking clear theoretical framing and given that developmental differences in hippocampal structure are not well established, the authors should test (establish) age differences in hippocampal anterior/posterior measures in this large sample. The authors include age as a covariate in all analyses. However, age-related effects should be of interest on its own (see comment about rationale above). Previous research has identified different sensitivity of the hippocampus to SES over age in developing

sample. With a big sample of wide age range, the authors are privileged to examine age-related effect on hippocampal subregions and cognition, differential SES effect on hippocampal subregions at different ages, e.g., potential age x SES interactive effect on the hippocampus. Moreover, these age-related effects may differ along the long-axis of the hippocampus, if the posterior hippocampus shows more protracted development. These analyses would provide more comprehensive picture of SES effect on hippocampal subregions across age and are crucial for drawing adequate interpretations.

We thank the reviewer for suggesting these analyses and we have included them in this revision. Thanks to your and Reviewer 1's comments, we explored how age interacts with our findings and have reported the results of these interactions in the text and supplement. We have also summarized the results of these analyses below.

Although both anterior and posterior hippocampal volume are positively related to age, posterior hippocampal volume had a stronger positive relationship with age than anterior hippocampal volume. We detail this important developmental finding in the Supplemental Materials on page 44 and report the estimates and statistics from the statistical model in Supplementary Table 36 (page 62) and visualize this finding in Supplementary Figure 6c (page 72).

Surprisingly, we found that age did not moderate income-cognition relationships, suggesting that income is positively related to cognition across development. We have now added this information to the text on pages 5-6, lines 101-106 and Supplementary Tables 37 and 38 and visualized this in Supplementary Figure 7a & b (page 73). Similarly, age did not moderate the relationship between income and anterior or posterior hippocampal volumes, suggesting that the positive effect of income on the anterior hippocampus is similar across children, adolescents, and young adults. This result is detailed on pages 9 of the text, lines 167-173, Supplementary Tables 39 & 40, and visualized in Supplementary Figures 8a and b.

In response to the Reviewer 1's fourth comment, we also tested whether age moderated cognition-volume relationships. As detailed above for the first reviewer, we found that age did not moderate anterior hippocampal volume-cognition relationships. However, age did moderate posterior hippocampal volume-cognition relationships. In particular, posterior hippocampal volume-cognition relationships were more positive among older than younger children. However, when we examined cognition-volume relationships in separate age bins (young children: 3-7, older children: 8-12, adolescents: 13-17; young adults: 18-21) there was no statistically significant relationships between posterior hippocampus and cognition in any age group. These findings suggest that posterior hippocampal does not reliably correlate with cognition in any age group. We noted this finding in the manuscript, page 11, lines 220-236, and detail analyses on discrete age groups in the supplement on pages 44-45, lines 948-975.

2. Stating limitation in the tasks used:

a. Language task seems to be merely about vocabulary. More broadly, it is unclear why

language measures are included in an investigation of hippocampal SES relations. Furthermore, the authors interpretation of the significant relation between language ability and anterior hippocampus is not clear.

This is an important point, and one that was also raised by Reviewer 1 (Comment 3). As we described in that response, the vocabulary test is calibrated to assess children's knowledge of low-frequency word meaning. Although the hippocampus is not classically associated with vocabulary, there is evidence that learning low frequency vocabulary depends on the hippocampus. In particular, hippocampal binding supports the ability to link a novel word to a new meaning or to a context of known words (James & MacKay, 2001; Davis & Gaskell, 2009; Duff & Brown-Schmidt, 2012). Functional MRI data also suggests that the hippocampus is related to novel word learning (Breitenstein, Jansen, Foerster, Wolbers & Knecht, 2005; David, Di Betta, Macdonald, Gaskell, 2009). Furthermore, individuals with hippocampal damage are impaired at learning new words and their meanings (vocabulary; Gabrieli, Cohen & Corkin, 1988; James & MacKay, 2001; Warren & Duff, 2014). This is what motivated us to look at this measure in association with the hippocampus; we have added these citations to the manuscript, page 2, line 39, page 4, line 70-71, as well as additional text justifying our choice to use a language measure on page 18, line 369-373: "*The hippocampus is also thought to contribute to novel word learning²⁶⁻³⁰, by associating novel words with a meaning or by extracting the meaning of a novel word from a broader semantic context²⁶⁻²⁹.*" And page 20, lines 413-416: "*the hippocampus is believed to be particularly involved in the acquisition and use of novel vocabulary²⁴⁻²⁸, making it a useful task for understanding how income differences in the hippocampus influence cognition.*"

b. Memory task is quite a short-term / working memory rather than an episodic memory task. Moreover, the authors don't provide a justification for the limitation of this specific task in representing 'memory' more broadly. This task has spatial recall component. What if this task characteristic, rather than a more general 'memory' process makes it selectively related to anterior Hc and not posterior hippocampus?

This is another important task consideration. The Picture Sequence Memory Test was designed to emulate episodic memory – by assessing memory for the spatial and temporal components of a temporally extended event – and its performance is strongly correlated with other standardized assessments of episodic memory (Dikmen et al, 2014). We agree, however, that the delay between image exposure and sequence recall is relatively short, and shorter than many episodic memory tests. Specifically, images are presented for 5 seconds each and sequence recall begins shortly after the last image. For the youngest children (who observe 6 pictures) this would result in a 30 second delay between the first image and recall. For the oldest (who observe 15 pictures) the delay would be over 1 minute. This is a longer delay than is typical for classic working memory assessments (Stone & Towse, 2015).

Importantly, the number of items presented to participants also exceeds limit of children's (Conway, Kane, Bunting, Hambrick, Wilhelm & Engle, 2005) and adults' (Cowan, 2010) working memory capacities, and participants encode and retrieve the sequence of images three times (Bauer et al., 2013), with improved performance on each repetition depending on long-term memory. For these reasons, we do not consider the task to be clearly a working memory task but do agree that working memory could contribute (alongside long-term memory) to performance on this task.

We have stated this directly on page 20 of the manuscript, lines 396-399, to alert the reader to this possibility: *"Although this task correlates with standardized measures of episodic memory^{63,74}, we note that there is only a short-delay between encoding and retrieval, and consequently, working memory processes may contribute to task performance."* We also note, though, that the ability to remember the spatial and temporal relationships is thought to depend on hippocampal binding, even over short delays (Olson et al., 2006; Olsen, Moses, Riggs, & Ryan, 2012) on page 20, line 399-401: *"Critically, however, the hippocampus is required for binding temporal and spatial relationships – even over short delays^{54,77} – making it an excellent task for assessing hippocampal function."* We now further motivate the use of this task in the manuscript on page 19, lines 377-379: *"The Picture Sequence Memory Test requires that participants associate an image with a particular temporal and spatial position and therefore is thought to rely on hippocampal binding⁵⁰."* and page 4, lines 69-71, *".. both memory and the acquisition and use of new vocabulary are thought to depend on hippocampal binding."*

The reviewer raises another important point here: with only one memory assessment, how do we know which aspect of the task is relevant to the mediation findings? Like any memory task, the Picture Sequence Memory Test does not assess all aspects of episodic memory, and without a battery of memory tasks, we agree that the relevant processes cannot be inferred. We have now stated in the manuscript that this task only measures some aspects of memory – spatial and temporal memory over relatively short delays, page 20, lines 401-403: *"Because this task targets short term memory for associations, the generalization of these findings to other aspects of memory – for example, long-term autobiographical memory – remains to be tested."*

We would also like to highlight that it is implausible that the posterior hippocampus would mediate income gaps in the performance of other memory tasks because income was not significantly related to posterior volume – a prerequisite for any income-cognition mediation analysis. Thus, while our design is not positioned to reveal the full scope of income-dependent cognitive functions which are mediated by the anterior hippocampus, the larger point that anatomical divisions of the hippocampus need to be considered to understand how it mediates income-cognition relationships still holds.

3. Missing information: On page 3, the authors mentioned “These model estimates demonstrate that memory ability in the poorest children appears to lag behind the wealthiest children by 4.5 years and language ability shows a similar lag of 5.5 years”. Please explain how the numbers of years were calculated. Regardless, authors should note the limitation of this interpretation based on cross-sectional study to the authors.

We thank the reviewer for finding this missing information. To calculate these numbers, we used the slope and intercept estimates of our models to determine how far apart the individuals in the lowest vs. highest income brackets were in their cognitive scores in z score units. We then divided the z score difference of their cognitive scores by the model estimate for age to rescale this difference in terms of years. We have now added a section describing in detail how we calculated this result and highlighted that the result should be interpreted with caution due to the cross-sectional sample, pages 45, lines 979-995. Additionally, this result is now visualized in Supplementary Figure 5, page 71 (also pictured below).

Supplementary Figure 5. Model estimates for the relationship between income and cognition revealed that a) memory in the lowest income bracket lagged behind the wealthiest by ~ 4.5 years, and b) vocabulary by ~ 5.5 years.

We have also noted in several places throughout the manuscript that our analyses are limited because of the cross-sectional design of our study, and that longitudinal data would allow for more reliable interpretations. We have stated this precaution after describing - not just for the analysis the reviewer is referring to – but also in relation to the reported mediation analyses. In Supplementary Materials page 46, lines 992-995 we state, “*We note that because of the cross-sectional nature of the study, these results should be interpreted with caution. Future research would benefit from using longitudinal data to measure the age-related trajectory of cognitive development and how memory differs in individuals from wealthy and low-income backgrounds.*”

For our mediation analyses, we also note the limitations of a cross-sectional sample on page 13, lines 262-265: “*Although a cross-sectional sample constrains our ability to infer for causal relationships, these findings are*

consistent with the possibility that income differences in the anterior hippocampus partially account for worse memory and vocabulary abilities among lower as compared to higher income individuals.” We also state in our discussion on page 15, lines 300-303: *“It is our hope that our correlative findings will inspire future longitudinal research that directly measures stress responses in children to determine whether stress can prospectively predict changes in anterior hippocampal volumes.”*

4. On page 9, the authors mention *“Notably, protective factors, such as cognitive stimulation, greater material resources, and better education, could boost hippocampal volumes in wealthier children, and thus play a role in the income-volume relationships that we observed. However, this possibility is inconsistent with our results, which showed that income-volume relationships were steepest at the lower end of the income distribution.”* As the relation between income and volume is linear (no significant quadratic effect), how is this ‘steepest’ conclusion drawn?

We thank the reviewer for highlighting that we omitted key information needed for interpreting our results. This is an important point that was also mentioned by Reviewer 1 (comment 2). In particular, we did not sufficiently emphasize in the manuscript that our analyses were fit using the log-transformation of family income, rather than raw income (in dollars). Thus, the linear relationships that we observe between the log of income and volume reflect the greater impact that income increments have on lower as compared to higher income groups. We chose to use the log transformation of family income to reflect the fact that a given increment of income to a lower-income family will have a greater impact than adding the same increment to a higher income family. Indeed, an additional \$5k annually has less impact on a family earning over \$200k annually than a family earning \$10k annually (Kahnemann & Deaton, 2010). However, doubling each family’s respective income may have a similar impact on both. The choice to use the log transformed income (rather than income in dollars) is meant to capture this idea, and is consistent with the methodology used in prior studies that examine how income influences measures of well-being and stress (Kahnemann & Deaton, 2010; Jebb, Diener & Oishi, 2018) as well as brain and cognitive health (Noble et al., 2012). Because of your and Reviewer 1’s helpful comments, we have clarified in the methods of our manuscript that the log of income was used in our models (see page 4 lines 79-87), and we also clarify this in the figure legends to highlight this important point for readers (see page 8 lines 148-154; page 10, lines 202-204). On page 4 lines 79-84 we state: *“Of note, the log transformation of family income was used in all analyses to reflect the idea, borne out in our data, that a given income increment to a lower income family has more impact than adding the same increment to a higher income family. Thus, in our analyses, the linear relationships reported reflect the greater impact that income increments have on lower as compared to higher income individuals.”*

Further, because of your comments, we now realize that we failed to conduct an important analysis that directly compared how steep income-volume and income-cognition relationships were in lower versus higher income groups. Based on your and the first reviewer's feedback we now more directly assess our claim by dividing our sample into two separate income brackets ($\leq \$75k$ vs. $> \$75k$). Notably, we observe that the relationship between income and anterior hippocampal volume is steeper among individuals in the lower half of the income distribution than the higher half. We report the results of these analyses in the manuscript on page 6, lines 107-122; page 9, lines 174-189; page 10, lines 204-207, page 11, line 234-236). The statistics and model estimates are also detailed in the supplement (See Supplementary Tables 20-35).

5. As for the measure of income:

a. First, why income? The authors decision to use income level as the only proxy of SES is not sufficiently motivated. Other measures, such as parental education (for enriched environment and parent-child interaction), social class or even composite score should also be considered.

We thank the reviewer for highlighting that we have not adequately clarified our rationale for using income rather than other proxies of SES for readers. Because our study was motivated through the lens of stress, we were interested in using a measure of SES that closely correlated with stress in the literature. In contrast to other measures of SES, such as parental education, there is a substantial literature documenting the close relationship between stress and income (Kahneman & Deaton, 2010; Jebb, Diener & Oishi, 2018; Felner, Brand, DuBois, Adan, Mulhall & Evans, 1995; McLoyd, 1998; Evans & English, 2002; Bradley & Corwyn, 2002; Evans & Cassels, 2013; Luby, Belden, Botteron, Marrus, Harm, Babb, Nishino & Barch, 2013). Because of this substantial literature, we felt that income was the most appropriate component of SES to test our hypothesis. We have clarified this in our paper on page 4, lines 74-78: "*We chose to use household income as our primary measure of SES because lower income is associated with greater stress*^{5,10,13,49-52} ..."; lines 88-89 "*We also report estimates and statistics from models that were fit using parental education in the supplement (Supplementary Tables 45-48)*".

Although we maintain that income is the most appropriate measure of SES for our study, we did have access to other measures of SES – namely data on parental education and occupation. Unfortunately, because we did not have control over data collection, we were limited to the variables available through the Pediatric Imaging Neurocognition and Genetics study and could not assess other measures of SES. Notably, within our sample, there was the highest number of data points available for parental income and it was the least skewed distribution; there were very few individuals in the two lowest parental education bins (e.g., $n = 4$ and $n = 6$; total n for parental education = 692) and fewer overall data points for parental occupation ($n = 653$). Therefore, in addition to our rationale for using income to infer stress

mechanisms, we thought that using parental income would allow for the most robust estimate of SES.

However, we have carefully considered the reviewer's point that parental education is also an important aspect of socioeconomic status. In response, we have now re-run our analyses using parental education as a predictor instead of parental income (See Supplementary Tables 45-48, Supplementary Figures 11-12). These analyses showed very similar results to that of income: parental education positively correlated with anterior hippocampus and measures of cognition, and anterior hippocampus mediated SES related gaps in language and marginally mediated SES gaps in memory. In addition, we found that parental education negatively correlated with posterior hippocampal volumes, though, posterior volumes did not mediate SES-gaps in cognition. We note that the different measures of SES are highly correlated with each other – parental income significantly correlates with both parental education $r(690) = 0.53, p < .001$ and occupation $r(651) = 0.54, p < .001$.

b. If income, then income-to-need ratio should be used instead, given that available resources to depends on the number of people in the household.

We agree that a measure of income-to-needs would be ideal. Unfortunately, however, data on the number of individuals in each household was not collected and therefore we are unable to calculate income-to-needs. We expect our use of income rather than an income-to-needs ratio has added noise to the relationships that we observed, rather than altered the direction of the relationships. Despite not having a measure of income-to-needs, our large sample affords high sensitivity, allowing us to reveal relationships with income – relationships which may be even more reliable with an income-to-needs ratio. We have now acknowledged in our manuscript that income-to-needs ratio has clear advantages over income, page 18, lines 362-365: *“Although an income-to-needs ratio would be an ideal measure of available resources, we did not have data on the number of individuals in each household and thus measures of income were used as our primary measure.”*

c. Are the SES of adults (18-21 yr) their own or their parents? If the latter, did the parents visit the lab? Please clarify the procedure.

We thank the reviewer for requesting this clarification. All household income data was for parents, even for individuals 18 and older. In the case that parents did not accompany participants who were 18 and older to the lab, participants were asked to report their parents' annual income. We have clarified this in the manuscript, on page 4, lines 71-73: *“Parental household income was also reported by accompanying parents or, in some cases, young adult participants”*. Excluding participants who are 18 years and older does not change the significant positive relationship between income and memory, income and language, or income and anterior hippocampal volumes.

6. More information should be provided on the selection of approach used for hippocampal segmentation. For example, were the 5 atlases used validated in developing sample and used T1 images as used here? Related, the authors should better clarify anterior and posterior segmentation boundaries.

We chose to use the Winterburn atlases in combination with MAGeT Brain algorithm because they have been validated in a sample of clinical and healthy adult participants on T1 images, similar to the images that we used in this study (Pipitone et al., 2014). Because a similar atlas developed by the same group has been validated in a developmental sample of neonates using T1 images (Guo et al., 2014) in conjunction with MAGeT brain, we expect these atlases to provide reliable segmentations of the hippocampus in our sample. This was confirmed by visual inspection of the data. Furthermore, a recent study found negligible differences in the reliability of subfield segmentations across child, adolescent and adult samples, even when segmentations are derived from adult atlases (Schlichting et al., 2018). This suggests that using adult atlases to segment the developing hippocampus results in labels that agree well with manually derived labels. In addition, since MAGeT registers atlases to a subset of the sample, 'the templates', and then uses the template labels to segment the entire dataset, labelling errors that might arise due to anatomical differences between the atlases and subject data are minimized. We have clarified this in the manuscript, page 22, lines 456-457. We also have clarified that we used the uncus to subdivide the anterior and posterior hippocampus, page 24, lines 482-489: *"To extract measures of the anterior and posterior hippocampus, a trained rater (A.L.D.) delineated the hippocampal labels into anterior and posterior segments. This was done by identifying the slice corresponding to the uncus apex, which is a commonly used landmark for the anterior-posterior hippocampus boundary⁸⁴, See Figure 2a for a visualization that marks the uncus apex."* And page 10, lines 196-199, *"An automated segmentation technique was used to segment the hippocampus and a manual segmentation approach was used to further subdivide the hippocampus into anterior and posterior segments at the uncus apex, marked by the dashed blue line."*

7. When conducting regressions for ICV correction across the whole sample, the assumption of homogeneity of regression should be checked. Given this is a sample with wide age range, the relation between ICV and ROI volumes may differ depending on age; if this is the case, the ICV correction should be conducted within sub-samples of different age. This may also be true for sex.

As per the reviewer's helpful suggestion, we tested whether the relationship between hippocampal subregions volumes and ICV differed at different ages and between the sexes, by splitting our sample into four age groups: younger children (3-7 years old), older children (8-12 years old), adolescents (13-17 years old), and young adults (18-21 years old), and using a regression to test for age * sex * ICV interactions. We found that age group moderated ICV-volume relationships, but there was no effect of sex. Therefore, we split our sample into different ages and corrected for ICV within

sub-samples of the data. These analyses and the sub-sample used for ICV correction are reported in the text, page 24-25, lines 502-516, and supplement, page 41-43. This methodological change, however, did not influence the pattern of results.

8. When conducting separate linear regression models, multiple comparison correction should be conducted to avoid increase rate of false positive, e.g., page 16.

We have now performed false discovery rate correction on all of our primary analyses and reported adjusted p-values that are corrected for the family-wise error rate. We have noted that we use FDR correction in the text, page 25, lines 522-525. *“We reported p-values that have been adjusted for the family wise false positive rate, using false discovery correction”*. The exploratory analyses suggested by reviewers are corrected for multiple comparisons using FDR separately from confirmatory tests. We note this on page 25 in the text, lines 524-527: *“Exploratory analyses suggested by the reviewers of this manuscript are reported in the text and supplement and were corrected for multiple comparisons separately from confirmatory tests. Results from exploratory analyses are detailed under the heading “Exploratory Analyses” in the supplement.”*

Minor points:

1. For figure 2 b, c, d, presenting individual data in scatterplots will greatly help the reader get a sense of the distribution and appreciate the findings depicted in the figures.

We thank this reviewer for this suggestion. After plotting the individual data for both anterior and posterior hippocampus on the same graph, we found it was difficult to visualize these relationships due to the large sample. Therefore, we have plotted these relationships - with the raw data - separately in the supplement, page 74, Figures 9 and 10.

2. The table of demographic can be moved to the main text. At least, information about age (mean, SD, range) should be mentioned at “participants” section.

We have moved the demographics information to the main text, page 5.

Reviewer #3

The authors find that the anterior hippocampus size, and not the posterior hippocampus, is partially mediating a relationship between income variation and memory and vocabulary abilities in children. While the relation between SES and cognitive achievement is well-established, this manuscript is an important advancement regarding the mixed findings in the hippocampus. The use of a well-powered dataset combined with an analysis that follows many best practices for data transparency and analysis is to be commended. The manuscript is extremely well written, reasonable, and

clear; it is a compelling work that will have substantial impact and be of use to multiple research areas in development and cognition. I have a few points for consideration, clarification or enhancement.

We thank the reviewer for these positive comments and are grateful that s/he thinks that our paper is compelling and informative.

1. Income is but one proxy for socioecological context. Often SES impact can be an accumulation of different risk factors, of which income is but one. Was race ever examined as a factor or controlled for in the analysis? While this study appears to use a majority white sample, perhaps at least minority status could be considered as a moderating factor of interest on the income-hippocampal-cognition relation.

We thank the reviewer for suggesting this analysis and have reported the results below. If the reviewer thinks these additional analyses are informative, we are enthusiastic about adding them to the paper.

Race and income-cognitive relationships: After adding minority status (white vs. all other) as a covariate to our analyses, income still significantly correlated with memory and language scores, all $ps < 0.05$). Memory and language scores were marginally lower among individuals with minority status relative to white individuals (Memory: $p = 0.06$; Vocabulary: $p < 0.001$). Adding minority status as a covariate *and* interaction term with income revealed no significant interaction ($ps > 0.34$) to predict cognitive scores, suggesting income-cognition relationships are similar in white and minority groups.

Race and income-anterior hpc volumes relationships: We also found that after adding minority status as a covariate, anterior hippocampal volumes still significantly correlated with income ($p = 0.04$). However, individuals with minority status had significantly smaller anterior hippocampal volumes than white individuals ($p < 0.001$). When we added minority status as a covariate *and* interaction term, we found a marginally significant interaction between race and income, $p = 0.053$, suggesting income-volume relationships were stronger among minority status individuals. Examining income-volume relationships separately for white and minority status individuals showed that income-volume relationships were significant among minority status individuals, but not white individuals. Plotting the data shows that individuals with minority status had smaller anterior volumes than white individuals, but only up to ~ 80k. See visualization below. This result may reflect additional stressors faced by individuals belonging to minority groups.

Race and income-posterior hpc volume relationships: There was a marginal relationship between income and posterior hippocampal volumes after adding minority status as a covariate ($p = 0.09$), but there was no effect of minority status on volumes ($p = 0.21$). Adding minority status as an interaction term revealed that there was no interaction between income and minority status.

Race as a covariate in mediation analyses: Finally, controlling for minority status in the mediation analysis did not substantially change our results: anterior hippocampal volumes marginally mediated income-memory, $p = 0.12$, and income-language relationships, $p = 0.09$.

Were any other SES factors such as parental education, single parenting, use of food stamps/SNAP, family mobility, zip code, or housing collected? Relatedly, as these data were collected from different scanners, and different parts of the country, quality of life can vary substantially for the same income (money can go much farther in Kansas than San Diego). How was this addressed in your sample? Some comments as to the limitations of using a single measure, such as income, as a proxy for SES should be added to the manuscript if other factors are not available to include in the model.

We have carefully considered the reviewer’s point that there are many factors that contribute to socioeconomic status. Unfortunately, there were only three measures of SES collected in the pediatric imaging neurocognition and genetics study: parental income, parental education and parental occupation. We chose parental income as our primary measure of SES because it has the strongest documented relationship with stress (Kahneman & Deaton, 2010; Jebb, Diener & Oishi, 2018; Felner, Brand, DuBois, Adan, Mulhall & Evans, 1995; McLoyd, 1998; Evans & English, 2002; Bradley & Corwyn, 2002; Evans & Cassels, 2013; Luby, Belden, Botteron, Marrus, Harm, Babb, Nishino & Barch, 2013). Furthermore, there were fewer data points for parental occupation ($n = 653$) and parental education ($n = 692$) particularly in the lowest education bins ($n = 4$ and 6). Therefore, we thought family income would provide the most reliable estimates of how different factors covary with SES. We note that all three measures of SES are highly correlated with each other –

parental income significantly correlates with parental education $r(690) = 0.53, p < .001$ and occupation $r(651) = 0.54, p < .001$.

To be as thorough and transparent as possible, however, we have now replicated our analyses with parental education and reported results from these analyses in the Supplementary Materials, page 67, Supplementary Tables 45-48, Supplementary Figures 11 and 12, to demonstrate that they do not depend on the particular proxy of SES used. These results showed the same pattern as income: parental education positively correlated with anterior hippocampus and measures of cognition, and anterior hippocampus mediated SES related gaps in language $ab = 0.02, SE = 0.017, 95\% CI [0.0002, 0.07]$ and marginally mediated SES gaps in memory, $ab = 0.005, SE = 0.003, 95\% CI [-0.0001, 0.01]$. Unlike income, parental education negatively correlated with posterior hippocampal volumes, though, posterior volumes did not mediate SES-gaps in cognition. We have added a statement in the manuscript directing readers to the analyses that use parental education instead of income, page 4, lines 87-89.

We also think that the reviewer has raised an excellent point that quality of life might vary for the same income in different cities. Because the data was collected in major US cities in which the cost of living is higher than that of the average cost of living in the US – San Diego, Los Angeles, Sacramento, Baltimore, New Haven, Boston, New York and Honolulu, we did not *directly* adjust income for the cost of living in each city. Instead, to control for potential differences, we included scanning site as a covariate in our analyses examining income-volume relationships and in our mediation analysis. We note that adding site as a covariate does not change our income-cognition relationships, and we are happy to add site as a covariate to the income-cognition analyses to control for site if the reviewer feels this is important.

2. Greater clarity is needed as to how “good parcellation” was determined, and “low movement” was determined from the scans. What was the “excessive” motion that resulted in removal, and how did it differ from acceptable motion?

Was a rating system or metric used for either type of QA (motion or parcellation)? If so, how were different quality levels determined? If not, how was consistency determined across scans, and how poor of quality was still acceptable? Similarly, on page 13, “a visual inspection was performed to ensure the quality of all images in the analysis”. The presence of what factors required removal from analysis, more specifically?

We are grateful to the reviewer for requesting these missing details. Each image was inspected for 1) motion rings and 2) ghosting outside of the image. Each image was assigned to 1 of 4 categories: no signs of motion detectable (1), minimal signs of motion detectable (2), clear signs of motion detectable (3), excessive motion detectable (4). Images that had either clear and excessive signs of motion were excluded from analysis ($n = 47$), whereas images with no signs of motion or minimal motion detectable were included ($n = 703$). Among the 703 scans included in the final analysis, 103 (15%) had minimal signs of motion (2), whereas 600 had no clear signs

of motion detectable (1). We note that prospective motion correction was used during the scan to minimize the effects of motion during data collection. We have added this information to page 21, lines 430-436 of the text.

We also inspected the hippocampal segmentations in coronal orientation to ensure the labels appropriately covered the hippocampus. Data was included if the segmentations covered the entire or the majority of the hippocampus on each slice that the hippocampus was visible. Otherwise, data was excluded. In total, two labels were excluded because the segmentations failed – and thus there was no label covering the hippocampus. There were no labels that extended beyond the boundaries of the hippocampus – which is consistent with prior findings that MAGeT Brain in conjunction with the Winterburn atlases tends to slightly underestimate, rather than overestimate the boundary of the hippocampus (Pipitone et al., 2014). To ensure consensus and consistency in our judgements, we had a second, blind rater successfully identify the 2 “failed” labels among a larger pool (n = 100) of accurate labels. We have added this information to the methods of the manuscript (page 23, lines 475-481)

We note that although there is some error in the labelling of the hippocampus, the volumes derived from MAGeT are well validated and correlate well with existing automated and manual techniques (Pipitone et al., 2014)

3. Minor points: Line 111 (page 6) refers to the wrong figure. Line 156 (page 8) is missing a “this” in “Consistent with possibility”.

Thank you for pointing this out. We have corrected these errors.

Reviewers' comments:

Reviewer #1 (Remarks to the Author):

The response to reviewers was thorough and well-researched. The authors did a terrific job running new analyses and fleshing out the data. Overall, there are assumptions about stress, and the appropriateness of the tasks for the questions asked, that remain problematic. I am still unconvinced by (1) the stress story and motivation. It remains pervasive in the paper. I think it is scientifically problematic to motivate, design a study around, and interpret findings based on an unmeasured variable. Lots of things vary by income, including nutritional status, access to materials resources etc. Again, a lot of assumptions. (2) the motivation of the vocabulary task is still weak. That the findings are inconsistent with Ghetti, whose episodic memory tasks have been functionally verified for construct validity as involving the hippocampal subregions, remains concerning.

Reviewer #2 (Remarks to the Author):

The authors have been highly responsive to the comments raised in my prior evaluation. The impressive amount of additional analyses and the adequate revisions toning down certain claims considerable strengthen the manuscript.

My remaining comments can be mostly addressed with minor revisions.

First, the effects of the two cognitive task scores are not differential. The patterns of findings of Income or volume and interactions are similar whether the picture sequence and picture vocabulary tests are considered. This leaves any statement of specificity of the effects somewhat questionable. The authors do not have data showing other cognitive test scores that would not show differential effects. Perhaps the use of IQ can substantiate the currently more specific claims about memory and language.

Second, I still see a need to point out the places where there was clear a priori rationale that fully motivated specific analyses, as may be the case for linking language specifically with the anterior hippocampus. These clarifications will help the reader better evaluate the findings that were not fully theoretically motivated.

Lastly, there are bits in the methods that need revision:

- 1) lines 429-435: description of data quality assurance fails to report the final sample size due to exclusion based on poor image quality.
- 2) lines 490-491: it matters if the same of differed by one slice...
- 3) line 480: not clear what larger pool refers to, which labels were off and in general how combining hyper segmented labels may influence differentially anterior and posterior subdivisions. How were labels (rather than the whole segmentation) used in segmenting the anterior from the posterior? hippocampus.

- 4) In addition (or instead) of Arndt et al. 1991 as a reference to using the common practice of ICV correction via an analysis of covariance, a reference should be added to Jack, et al 1989.
- 5) Supp figure 6c title is mislabeled.

Reviewer #3 (Remarks to the Author):

I appreciated the authors strong additions to the manuscript and thoughtful responses to the reviewers' concerns. I think the paper is much stronger as a result of their additions and edits. I have no more substantial concerns, though a lingering concern about the minority status results that the authors produced in response to my previous comment.

- It gave me great pause to see that minority status drove the anterior hippocampus correlation with income (the primary result, and the figure on page 21 of the rebuttal). The effect is not seen in white individuals alone. The substantially smaller hippocampal volumes in minority individuals also led me to wonder how age interacted with minority status. I cannot determine from the demographic tables how many individuals <\$75K are minority status vs. white, but the much smaller power in analyses of minority individuals (given their overall small number) make me uncertain as to whether to include this result, and will defer to the editorial team. Overall, the primary drivers being the minority status individuals of the anterior hippocampus effect adds nuance to the overall argument of the paper, and calls for more diverse future public samples, as a few poor, minority individuals seem to be having a strong effect on the current correlations of brain structure and income.
- The addition of scanner site as a covariate to the analyses is sufficient as is.
- Given the large number of supplemental tables currently, I would appreciate a 1-2 sentence text summary of findings after each question presented in the Exploratory analyses.
- The greater methods description as to quality control is appreciated.
- Supplemental Figure 6 has the text description of the result reversed (age has a larger effect on POSTERIOR volumes)
- Page 3, line 64: age range is misreported

Response to reviewers' comments

Reviewer #1 (Remarks to the Author):

The response to reviewers was thorough and well-researched. The authors did a terrific job running new analyses and fleshing out the data. Overall, there are assumptions about stress, and the appropriateness of the tasks for the questions asked, that remain problematic. I am still unconvinced by (1) the stress story and motivation. It remains pervasive in the paper. I think it is scientifically problematic to motivate, design a study around, and interpret findings based on an unmeasured variable. Lots of things vary by income, including nutritional status, access to materials resources etc. Again, a lot of assumptions. (2) the motivation of the vocabulary task is still weak. That the findings are inconsistent with Ghetti, whose episodic memory tasks have been functionally verified for construct validity as involving the hippocampal subregions, remains concerning.

Thank you for this additional feedback. We take the reviewer's concern about stress not being measured seriously. Before describing how we addressed this concern, we would like to reiterate why we feel it is important to highlight stress when motivating our question and methodological approach. Unlike other unmeasured variables, stress has been emphasized by major theories to explain why low SES individuals have smaller HPC volumes (See Farah, 2008; Hackman & Farah, 2009; Hackman, Farah, Meaney, 2010; Farah 2017, Farah 2018). Therefore, in conducting a study about smaller HPC volumes in low SES individuals and how the hippocampus might mediate these cognitive gaps, we felt it was appropriate to place our work within the context. Moreover, our methodological decision to divide the hippocampus into anterior and posterior divisions was motivated by the stress literature.

We have therefore kept stress as a *possible* mechanism in the paper while acknowledging other variables that are associated with low income. Specifically, we have maintained the hippocampus has been singled out by predominant theories because of its vulnerability to stress and because of the relationship between SES and stress – as we feel it would be inappropriate to ignore this work that motivated the study. However, we have made several additional changes to emphasize that other variables may also play a role.

For example, in the first paragraph of the manuscript on **page 3, lines 31-35**, we state: *“These cognitive gaps coincide with differences in brain structure that are thought to reflect the cumulative effects of income-related stress^{2,2,5-8} and environmental stimulation on brain development^{2,9,10}. Indeed, lower income is associated with fewer material and nonmaterial resources^{2,9,10} and a higher number of stressful life events^{5,11-16}...”*

Along these lines, we agree very much that it is very important to highlight that other factors covary with income and that we did not measure stress directly. We emphasize this throughout the manuscript.

On **page 19, line 352-356** – Right after summarizing our findings in the Discussion, we state: *“The large public dataset used here did not include a stress measure, which limited our ability to directly assess the role that income-related stress played in our findings. Thus, stress remains a plausible mechanism, but not the only possible one. Indeed, many other factors, including nutrition⁷⁶ and access to material and non-material resources^{2,9,10} correlate with socioeconomic status.”* We also highlight in numerous places throughout the manuscript when interpreting our findings that stress is an unmeasured variable, and that our data is consistent with the possibility that stress underlies our findings, but that future work needs to be done that directly measures stress. For example, on **page 19 line 365-368**, we state: *“It is our hope that our correlative findings will inspire future longitudinal research that directly measures stress responses to determine whether stress can prospectively predict changes in anterior hippocampal volumes during development.”*

Thank you also for commenting on the vocabulary measure. We have added further justification for our decision to include the vocabulary task in our study on **page 5 lines 79-83**, emphasizing that hippocampal damage impairs new word learning, and that both hippocampal structure and function correlate with vocabulary across the lifespan. We state: *“Indeed, individuals with hippocampal damage are impaired at learning the meaning of new words^{62,66-68}, and numerous studies demonstrate that language acquisition and vocabulary abilities correlate with hippocampal activity^{29,32,69}, and hippocampal volume in infants⁷⁰, children⁷¹ and adults⁷²..”* Given the evidence in the literature linking the hippocampus to word learning, and low SES to poorer language abilities, we think it is important to include this measure of vocabulary in our study.

Last, we acknowledge that these findings differ from those which were reported in the Demaster & Ghetti (2013) paper. Because our work was not designed as a replication of Ghetti (2013), we are not surprised that these findings differ from theirs. There are many differences between the two studies. For example, Demaster & Ghetti divided the hippocampus into 3 subdivisions, whereas we divided the hippocampus into 2.

We think that the memory task that we included is unlikely to be the cause of these differences because our results are consistent with two other studies that assess both spatial-temporal memory in adults and children’s episodic memories. In particular, Rajah, M. N., Kromas, M., Han, J. E., & Pruessner, J. C. (2010) showed a positive relationship between anterior hippocampal volumes and spatial and temporal context memory in young adults. Additionally, Riggins, Blankenship, Mulligan, Rice, Redcay, 2015 showed that in children (6-year olds), anterior but not posterior hippocampal volumes positively correlate with episodic memory. We now cite this research in the manuscript on **page 14 line 264**. We also note that the validation work done on the memory task that we used has shown that performance on the task correlates with other episodic memory measures (Bauer 2013) which we now highlight on **page 25, lines 485-486**: *“However, validation studies of this task have shown that this task has sufficient*

construct validity and correlates well with performance on other measures of memory⁸⁵.”.

Reviewer #2 (Remarks to the Author):

The authors have been highly responsive to the comments raised in my prior evaluation. The impressive amount of additional analyses and the adequate revisions toning down certain claims considerable strengthen the manuscript.

My remaining comments can be mostly addressed with minor revisions.

We are glad that the reviewer thinks the previous revisions have strengthened the manuscript and have now addressed the reviewer’s remaining comments.

First, the effects of the two cognitive task scores are not differential. The patterns of findings of Income or volume and interactions are similar whether the picture sequence and picture vocabulary tests are considered. This leaves any statement of specificity of the effects somewhat questionable. The authors do not have data showing other cognitive test scores that would not show differential effects. Perhaps the use of IQ can substantiate the currently more specific claims about memory and language.

Thank you for this suggestion. We do not have IQ measures, but do have measures of performance on the flanker task of inhibitory control which is thought to be independent of the HPC. To address your concern, we have now used this measure as a control task to assess whether income and HPC volumes correlate with HPC-independent cognitive processes, or whether these relationships are selectively related to hippocampal-dependent processes (i.e., memory and vocabulary). Since income is thought to broadly correlate with cognition, we expected income to correlate with both HPC-dependent and independent processes, but we expected that HPC volumes would be selectively related to HPC dependent processes.

We found that income did correlate with inhibitory control abilities ($p = 0.04$), albeit to a lesser extent than episodic memory and vocabulary. The effect size for income-inhibitory control relationships was 2.5 times smaller than that for income-episodic memory relationships, and 4 times smaller than that for income-vocabulary relationships ($d = 0.16$ for inhibitory control; $d = 0.41$ for memory; $d = 0.68$ for vocabulary). Moreover, an interaction analysis showed that income-memory and income-vocabulary relationships were significantly stronger than the relationship between income and inhibitory control. Together, these findings suggest while income has broad effects on cognition, it most strongly correlates with hippocampal-mediated processes. This result is consistent with prior research (Farah et al., 2006).

Interestingly, we also found that while memory and vocabulary correlated with anterior HPC volumes, inhibitory control did *not* correlate with either anterior or posterior HPC volumes (p s > 0.18). Moreover, results from an interaction model showed that vocabulary had a significantly stronger relationship to anterior HPC volumes than inhibitory control; though, memory did not have a significantly stronger relationship with anterior HPC volumes than inhibitory control. Taken together, these findings suggest the anterior HPC correlates only with aspects of cognition known to be supported by the HPC. However, the relationship with anterior HPC volume is not significantly different for memory and inhibitory control. We note that we also had a measure of processing speed and tested whether processing speed related to income and HPC volumes to confirm that HPC volumes were only associated with HPC-dependent tasks, though, for the sake of simplicity we did not include these findings on processing speed in the manuscript. However, we note that similar to the pattern for inhibitory control, we found that processing speed correlated with income ($p = 0.008$) but did not correlate with anterior nor posterior HPC volumes (p s > 0.14).

We have now added the flanker task as a control task to our manuscript and motivate its inclusion in the introduction. On **page 4, lines 79-83**, we state: *“Of note, we also tested whether hippocampal subregion volumes were related to performance on a flanker task of inhibitory control. This task was used as a control measure, allowing us to determine whether volume-cognition relationships were specific to memory and language (known to be supported by the hippocampus) or related to other aspects of cognition (inhibitory control) that are thought to be hippocampal-independent.”*

We now report this result in the body of the manuscript. On **page 7, lines 113-123** we state: *“We also observed a positive relationship between family income and inhibitory control ($b = 0.10$, $SE = 0.05$, $t(678) = 2.11$, $p = .035$, p -adjusted = 0.035; $r = 0.08$), suggesting, as other studies have¹, that income has broad effects on cognition beyond cognitive processes supported by the hippocampus. However, an interaction analysis revealed that income was more strongly correlated with memory ($b = 0.09$, $SE = 0.03$, $t(1370) = 2.95$, $p = .003$, $r = 0.08$) and vocabulary ($b = 0.14$, $SE = 0.03$, $t(1370) = 4.25$, $p < .001$, $r = 0.11$) than inhibitory control. Indeed, the effect size of the relationship between income and memory ($d = 0.41$) and language ($d = 0.68$) were 2.5 times and 4 times larger than the effect size for inhibitory control ($d = 0.16$). Together, this suggests that income has more prominent effects on hippocampal-dependent processes—a result that is consistent with prior work¹.*

Moreover, on **page 8, lines 147-150**, we state: *“In contrast, there were no differences in income-inhibitory control relationships in the lower versus higher income group (interaction: $b = -0.000002$, $SE = 0.000001$, $t(676) = -1.56$, $p = .119$, $r = 0.06$) suggesting these relationships were similar in higher and lower income groups.”*

We have also reported that hippocampal volumes do not correlate with inhibitory control. On **page 14-15, lines 271-280**, we state: “As expected, neither anterior nor posterior hippocampal volumes correlated with inhibitory control (anterior: $b = 0.0002$, $SE = 0.0001$, $t(677) = 1.32$, $p = .186$, $r = 0.05$; posterior: $b = 0.00009$, $SE = 0.0002$, $t(677) = 0.53$, $p = .596$, $r = 0.02$). Moreover, interaction analyses showed that anterior hippocampus was marginally more correlated with vocabulary than inhibitory control ($b = 0.0002$, $SE = 0.0001$, $t(1371) = 1.83$, $p = .068$, $r = 0.05$), though not more correlated with memory than inhibitory control ($b = 0.00008$, $SE = 0.0001$, $t(1371) = 0.79$, $p = .430$, $r = 0.02$). Thus, while anterior hippocampal volumes only correlated with cognitive processes known to depend on the hippocampus, the relationships between anterior hippocampal volumes and memory and inhibitory control did not differ significantly from each other.”

We have also described the flanker task in the Methods section (**page 26-27**).

Second, I still see a need to point out the places where there was clear a priori rationale that fully motivated specific analyses, as may be the case for linking language specifically with the anterior hippocampus. These clarifications will help the reader better evaluate the findings that were not fully theoretically motivated.

Thank you. We have now stated our a priori hypotheses in the text. We hypothesized a priori that income would correlate with memory and language scores on **page 7, lines 107-108**: “We first confirmed our a priori hypothesis that family income was positively related to episodic memory and vocabulary scores across the sample”.

We also had an a priori hypothesis that there would be a stronger relationship between income and the anterior hippocampus than income and the posterior hippocampus, and that if this were the case, anterior hippocampus would mediate income-related gaps in cognition. We now state this in the text on **page 4, lines 65-69**: “Based on this work, we hypothesized that anterior hippocampal volumes would be disproportionately smaller in low than high income children relative to the posterior hippocampus. Moreover, we expected that if this were the case, anterior hippocampus would mediate income-related gaps in cognitive abilities that are supported by the hippocampus.”

In contrast, we did not hypothesize that cognitive scores would be selectively related to the anterior hippocampus. We now state this on **page 14, lines 261-268**: “We next tested whether episodic memory and vocabulary were related to the anterior and posterior hippocampus. Although we did not have a priori predictions that these cognitive tasks would be selectively related to a specific hippocampal subregion, we found that episodic memory and vocabulary were positively related to anterior but not posterior hippocampal volumes.”

We have now also clarified that the analyses suggested by the reviewers are exploratory and not part of our original hypotheses and thus we did not have a priori predictions for these questions. On **page 5, lines 73-74**, we state: *“We also performed exploratory analyses to test whether age moderated our results”*, and on **page 6, lines 98-101**: *“While these analyses were exploratory, we expected that linear gains in income would correspond to larger increases in cognitive scores and hippocampal volumes in the lower (\leq \$75k) than higher income subsample ($>$ \$75k)”*

Lastly, there are bits in the methods that need revision:

1) lines 429-435: description of data quality assurance fails to report the final sample size due to exclusion based on poor image quality.

Thank you for pointing this out. We now include the sample size leftover after excluding individuals based on poor image quality on **page 28 lines 546-548**. *“In total, 38 participants were excluded at this stage for clear and excessive signs of motion, leaving 712 eligible participants.”*

2) lines 490-491: it matters if the same of differed by one slice...

We have now clarified that the raters identified the same slice in 94% of cases, and the same or a slice that differed by 1 slice in 98% of cases. We adapted this benchmark from the deMaster and Ghetti 2014 paper who reported inter-rater reliability for the boundary by slices that were the same or differed by 3 slices or less. On **page 30, lines 605-607**, we state: *“For the anterior-posterior boundary, the researchers identified the same slice in 94% of cases, and the same or a slice that differed by 1 slice in 98% of cases.”*

3) line 480: not clear what larger pool refers to which labels were off and in general how combining hyper segmented labels may influence differentially anterior and posterior subdivisions. How were labels (rather than the whole segmentation) used in segmenting the anterior from the posterior hippocampus?

We have now clarified that to ensure objectivity in excluding 2 of the labels, we selected a random subset of 100 images from our sample (including the two ones deemed poor quality), and had a blind rater identify the two poor quality labels (and only those two labels).

We now also clarify that on these hippocampal slices, the labels covered only a few (3 or 4) slices of the hippocampus. On **page 30, lines 589-595**, we state: *“In the case of 2 labels, the segmentations only covered a few (3 or 4) slices hippocampus. These 2 labels were excluded, leaving a total of 710 labels. To ensure objectivity for excluding these two labels, we selected a random subset of 100 labels among the 712 MR images (including the two labels that we decided to exclude due to poor segmentation). We then had a second blind rater successfully identify these 2 poor quality labels (and only these labels) for exclusion.”*

We have also clarified that the divisions between subfield labels were not used in any way to inform the division between the anterior and posterior hippocampal boundary. We now state on **page 30, lines 608-616**: *“After identifying the boundary slice, any part of a subfield label (CA1, CA2/3, CA4/DG, subiculum, SRLM) that fell rostral to the uncus was counted towards the volume of the anterior hippocampus, whereas any part of a label that fell caudal to the uncus was counted towards the volume of the posterior hippocampus. Thus, the volume of the anterior and posterior segments, respectively, reflected the total voxels covered by any subfield label that was rostral or caudal to the boundary slice. In this way, the subfield labels were largely treated as though they were a single label: we ignored the divisions between them, and they were not used to inform the boundary between anterior and posterior segments.”*

4) *In addition (or instead) of Arndt et al. 1991 as a reference to using the common practice of ICV correction via an analysis of covariance, a reference should be added to Jack, et al 1989.*

We have now cited Jack (1989) instead of Arndt (1991).

5) *Supp figure 6c title is mislabeled.*

Thank you. We have corrected the title of the figure.

Reviewer #3 (Remarks to the Author):

I appreciated the authors strong additions to the manuscript and thoughtful responses to the reviewers' concerns. I think the paper is much stronger as a result of their additions and edits. I have no more substantial concerns, though a lingering concern about the minority status results that the authors produced in response to my previous comment.

We appreciate the reviewer's feedback and also feel that the manuscript is much stronger as a result.

It gave me great pause to see that minority status drove the anterior hippocampus correlation with income (the primary result, and the figure on page 21 of the rebuttal). The effect is not seen in white individuals alone. The substantially smaller hippocampal volumes in minority individuals also led me to wonder how age interacted with minority status. I cannot determine from the demographic tables how many individuals <\$75K are minority status vs. white, but the much smaller power in analyses of minority individuals (given their overall small number) make me uncertain as to whether to include this result and will defer to the editorial team. Overall, the primary drivers being the minority status individuals of the anterior hippocampus effect adds nuance to the

overall argument of the paper, and calls for more diverse future public samples, as a few poor, minority individuals seem to be having a strong effect on the current correlations of brain structure and income.

Thank you for your comment. Understanding how minority status moderates these relationships is an important question, and one that has not received enough attention in the literature. Although this was not our initial motivation for conducting the study, we think that our data might provide preliminary insights, though not conclusive answers to this question. One reason why our analyses should be considered as strictly provisional, as you note, is that the distribution of whites (non-minority status individuals) and non-whites (minority status individuals) is far from evenly dispersed across income bins, especially in the higher income half of the sample (>75k), where there are only 40 minority status and 253 non-minority status individuals. In response to your thoughts, however, we have now reported full details of our analyses related to minority status in the supplement, and refer to the findings in the results section of the paper in the appropriate section.

In recognition of the highly skewed distribution of income across minority groups, we restricted analyses probing effects this factor to individuals who earn 75k and less. Fortunately, this lower income group is of greatest theoretical interest because stress correlates with income only to an annual income of 75k (Kahnman & Deaton 2010). This choice also aligns with our findings that anterior hippocampal volumes correlate with income in the lower, but not higher income subsample, and with Reviewer's 1's comment in the last revision that income is not a meaningful predictor of stress among wealthy individuals only. Within the lower income subsample ($\leq 75k$) we have a larger sample of minority status individuals ($n = 134$ minority status, $n = 268$ non-minority status), boosting our power to detect real effects. However, even within this lower income sample, the proportion of minority status individuals is much greater in the lower as compared to higher end of the income distribution. In fact, even within the lower income group, there are nearly 3x the number of minority status individuals in the lowest income bin and 3x the number of non-minority status individuals in the highest income bin (See figure below).

With these caveats in mind, we report findings from (1) analyses that test how minority status moderates income-volume and (2) income-cognition relationships and have also (3) detailed results split by minority status in the supplement. We have also included supplemental tables detailing how age interacts with minority status to predict anterior hippocampal volumes, as per the reviewer's request, though the age x minority status interaction was not reliable (**See supplementary Tables 60-61, page 82-83**).

To summarize our results in the lower income sample ($\leq 75k$):

- (1) minority status did not significantly moderate the relationship between income and cognition ($p_s > 0.26$); further analyses showed income correlated with episodic memory and vocabulary in both non-minority status and minority status individuals (all $p_s < 0.05$; **See supplementary Tables 49-54 on page 76-79**).
- (2) Minority status also did not significantly moderate the relationship between income and anterior or posterior hippocampal volumes ($p = 0.20$); however, as in the analyses that included the full sample, income-anterior volume relationships were only significant in minority status ($p = 0.018$), but not non-minority status ($p = 0.37$), see **supplementary Tables 55-58, pages 79-81**.

We agree that the fact that income correlates with anterior HPC volumes in minority status but not non-minority status individuals certainly adds nuance and calls for more research on this topic. However, given the lack of a significant interaction between minority status and income, we do not want to draw overly strong conclusions based on this result. Indeed, the non-significant interaction indicates that the relationship between income and anterior hippocampal volume is not significantly different between non-minority status and minority status individuals and prior researchers (Nieuwenhuis, Forstmann, Wagenmakers (2011) have cautioned against dividing data when there is a nonsignificant interaction and this may be especially important when such analyses were not planned (as is the case here). Further complicating matters, the null effect observed in the non-minority status group might reflect the relatively high incomes in this group: 54% of non-minority status individuals are in the 75K income bin. By comparison, the minority status group has more evenly distributed incomes and are, thus, better suited to income analyses. The fact that we expect the effects of income to be most pronounced in the lower income bins also may explain why we see this effect in minority status, but not non-minority status individuals.

Although we agree that these findings are noteworthy (and we are reporting all the results), it is reassuring for the generalizability of the findings that income remains a significant predictor of anterior hippocampal volumes ($p = 0.009$) even in models that include minority status as a covariate and interaction term. The fact that even after accounting/controlling for minority status in the model, income remains a significant predictor, illustrates that these findings are generalizable (at least according to the statistical model). Therefore, at this stage, we feel it would be premature to conclude that these results only apply only to non-white individuals.

To address the topic of minority status in the paper, we now include a discussion paragraph about the limitations of drawing conclusions about minority status using our sample, as well as about the generalizability of our results to non-minority status individuals (how including minority status in the models does not eliminate the relationship between income and anterior hippocampal volumes) (**Page 20, lines 384-399**). We have also used this paragraph as an opportunity to call for more diverse samples in studies of brain development, as many studies examining brain development limit their samples to children of white educated parents.

Of note we refer to the minority status analyses in the results section (minority status effects on income – cognition relationships: **page 9, lines 154-158**; income-volume relationships **page 13, lines 231-239**), and described the statistical models in the methods section (**page 34, lines 678-688**; **page 35, 715-720**)

The addition of scanner site as a covariate to the analyses is sufficient as is.

Thank you.

Given the large number of supplemental tables currently, I would appreciate a 1-2 sentence text summary of findings after each question presented in the Exploratory analyses.

We have now added a 1-2 sentence summary of our findings after supplementary table in the exploratory analyses section.

The greater methods description as to quality control is appreciated.

We are happy that the reviewer is satisfied with the description of quality control.

Supplemental Figure 6 has the text description of the result reversed (age has a larger effect on POSTERIOR volumes)

Thank you. We have corrected the title of the figure.

Page 3, line 64: age range is misreported

Thank you for pointing this out. We have now corrected the age range.

Reviewers' comments:

Reviewer #2 (Remarks to the Author):

The addition of inhibitory control task contributes to strengthening the manuscript. However, it should be more explicitly reported including scatter plots so that the variance in the measure can be evaluated. For example, in analyzing the data from the flanker task, the authors seem to have used different measures for high performers (>80% correct) vs low performers, this introduced variance not equally across the sample. How would the results look if just using accuracy (which may be skewed due to ceiling effect)?

I remain concerned with the authors equating performance on a task with an individual ability on cognitive construct that the task is thought to be associated with. It would be better, especially when analyses are reported, if the authors describe scores of tasks, and leave referencing cognitive constructs to the discussion. Related, I find the analysis pitting tasks against each other in assessing relationship with income not meaningful at all. Particularly without more information about test for normality and other screening that may alleviate challenges to this analysis. Given tasks not only measure different alleged constructs but each have unique stimuli and characteristics making interpretations of differences between scores inherently very limited. Further concern is that for the vocabulary test, difficulty seem to be adjusted differently for when tested, it is not clear to me how does this influence the measures and whether it still reasonable to z-score such 'raw' scores when assessing individual differences.

Similar to Reviewer 1, I also remain somewhat skeptical by the framing of stress without strong theoretical motivation and find it difficult to make much of differential correlations in anterior vs posterior hippocampus. I understand that the authors are committed to this perspective however, this remains problematic. I always worry that when theoretically unsupported findings make a good story, we inevitably move further away from a chance to uncover real principles.

Remaining concerns regarding analyses of brain volumes:

1. On page 28, line 577, citation 96 tested ANTS and ASHS, and thus is not an appropriate citation to make the point that adults atlas applies in children sample as the authors are using MAGeT in current study.
2. On page 31, line 639-641, before combining the left and right volumetric measures, the hemisphere-related effects (e.g., age x hemisphere, SES x hemisphere) in predicting volume should be tested. Only combine if the interactions are not significant.

Remaining concern regarding measures of SES: please add limitation of not using income-to-needs ratio in discussion.

Remaining concern regarding treating age: given the large age range non-linear trends need to be tested consistently. Handling of age is still not sufficiently described and non-linear terms are only adopted in

one analysis (described starting in line 698), yet, due to the large age range (and good theoretical, and practical reasons) should be adopted consistently.

Minor:

1. When reporting results, explain what r value is for the first time (Page 7, line 110).
2. On page 8, line 146, figures 1e & f are mislabeled and should be f&g.
3. On page 9, line 158, supplementary tables 55-58 numbers are mislabeled and should be 49-54.
4. For figure 4, add explanation for the value in parentheses.
5. Supplementary Table 36, was the regions (anterior/posterior) entered as repeated measures?

Reviewer #3 (Remarks to the Author):

I find the authors' response thoughtful and their points valid. I have no further concerns.

Reviewers' comments:

Reviewer #2 (Remarks to the Author):

The addition of inhibitory control task contributes to strengthening the manuscript. However, it should be more explicitly reported including scatter plots so that the variance in the measure can be evaluated. For example, in analyzing the data from the flanker task, the authors seem to have used different measures for high performers (>80% correct) vs low performers, this introduced variance not equally across the sample. How would the results look if just using accuracy (which may be skewed due to ceiling effect)?

Thank you for this question. Yes, the flanker scores were adjusted differently for those who scored above versus below 80% correct, in keeping with validation studies of the NIH toolbox flanker task (Zelazo et al., 2013, Zelazo et al., 2014). We do see your point, though, that this approach could distort individual differences. Because this open access database only provides a few summary statistics characterizing participants' flanker task performance, our options are limited. Based on your concern, we considered using incongruent trial accuracy as an alternative. However, incongruent trial accuracy is highly skewed due to ceiling effects (See distribution below). Therefore, we decided not to use this as a control measure. Instead, we revised the manuscript using a different construct and associated measure—processing speed—which is not hypothesized to relate to hippocampal volumes, and which also does not include a performance-based adjustment.

To assess processing speed, participants were shown images, side by side, on a computer screen, and were asked to determine whether the images were the same or different. The pictures were simple drawings depicting common things (e.g., clouds, trees), and were either the same or differed on one of three

dimensions: (1) color; (2) adding/taking something away; (3) one versus many; see depiction of example images below from Carlozzi et al., (2015). Scores on the task reflect the number of correct items that participants completed in 90 seconds.

Similar to the flanker index results, we found that income has an expected correlation with processing speed ($p = .008$, $r = .10$; **page 7, lines 116-118**). Interestingly, unlike vocabulary and memory scores, there were no differences in the relationship between income and processing speed in the lower and higher income subsample ($p = .67$, $r = 0.02$; **page 8, lines 144-145**) indicating that the relationship between processing speed and income is present in both lower and higher income samples. As predicted, we also found that processing speed scores were unrelated to anterior or posterior hippocampal volumes (all p s > 0.05 ; **page 13, lines 262-266**). In line with your suggestion, we also added scatter plots so the readers can fully evaluate the variance in processing speed scores (**Supplementary Figures 13a and b on page 93**).

Supplementary Figure 13 a) Family income was significantly correlated with processing speed ($p = 0.008$) after controlling for age and sex. *b)* Anterior hippocampal volumes did not correlate with processing speed after controlling for age and sex ($p = 0.13$). In both figures, residualized values for processing speed scores are plotted after removing variance associated with age and sex to better visualize the relationships tested. Grey shading reflects standard errors.

I remain concerned with the authors equating performance on a task with an individual ability on cognitive construct that the task is thought to be associated with. It would be better, especially when analyses are reported, if the authors describe scores of tasks, and leave referencing cognitive constructs to the discussion.

Thank you for this suggestion. When reporting results from cognitive tests, we now refer to “scores” (e.g., memory scores, vocabulary scores).

Related, I find the analysis pitting tasks against each other in assessing relationship with income not meaningful at all. Particularly without more information about test for normality and other screening that may alleviate challenges to this analysis. Given tasks not only measure different alleged constructs but each have unique stimuli and characteristics making interpretations of differences between scores inherently very limited. Further concern is that for the vocabulary test, difficulty seem to be adjusted differently for when tested, it is not clear to me how does this influence the measures and whether it still reasonable to z-score such 'raw' scores when assessing individual differences.

Thank you for this feedback. The aim of this analysis was to assess whether income was *more strongly* correlated with vocabulary and episodic memory scores than another control task that has not previously been associated with the hippocampus (here, this is now processing speed). In the last version of the manuscript, when assessing the difference in these scores' relationships with income, we chose to z-score the raw scores to ensure that all of the scores could be compared to each other on the same scale, decreasing the possibility that differences in each of the scores original units influenced potential interaction effects. Notably, the vocabulary scores are on a much smaller scale than the processing speed and memory scores, making it likely that the slope in its relationship with income could differ due to the scale of the tasks' original units.

Importantly though, we understand your concern that a difference in the relationships of these scores with income does not necessarily reflect a difference in the relationship between income and the *cognitive constructs* that the tasks are designed to measure. Therefore, we have removed analyses pitting tasks against each other from the manuscript (though we have re-run the analysis for you here). If the reviewer finds it useful for us to include this analysis, to determine whether income is more strongly related to vocabulary and episodic memory than the control task, we are happy to put this back in the paper.

To summarize the results for processing speed (since in the previous version we included an inhibitory control task), linear mixed effects modelling (*i.e.*, $lmer(scores \sim income (log) * task + age + sex + (1|participant))$) showed that income was more strongly related to memory ($p = .009$, $r = 0.07$) and vocabulary ($p < .001$, $r = 0.10$) than processing speed. Of note, a Kolmogorov-Smirnov test

for normality demonstrated that the residuals of the model were normally distributed.

Similar to Reviewer 1, I also remain somewhat skeptical by the framing of stress without strong theoretical motivation and find it difficult to make much of differential correlations in anterior vs posterior hippocampus. I understand that the authors are committed to this perspective however, this remains problematic. I always worry that when theoretically unsupported findings make a good story, we inevitably move further away from a chance to uncover real principles.

Thank you. The theories about SES and stress and the research in animal models of stress (that we outline in our introduction) motivated our hypotheses. But we agree that stress is not the only factor that could influence our findings. In response to this feedback, we have de-emphasized stress as a primary potential mechanism when discussing the possible mechanisms that mediate our findings. In line with this, and in conversations and advice from the editor, we have now amended our manuscript in several ways.

- (1) We have reframed the abstract. Although stress remains a motivation behind the work, we removed it from the interpretation of our results and conclusions. In the abstract, rather than conclude that low income children could have a dysregulated stress response, we now conclude by stating: *“Our findings add much needed anatomical specificity to current theories by showing that family income disproportionately affects the anterior hippocampus.”*
- (2) We have now stated in our introduction that we do not have a direct measure of stress to alert readers to the fact that we didn’t measure stress and therefore cannot infer that stress mediated the effects we observed. On **page 5, starting on line 76**, we state *“Importantly, although our hypotheses are inspired by the animal work described above on the impact of chronic stress on the anterior hippocampus, we do not have a direct measure of stress in our dataset and are therefore unable to directly test the impact of stress specifically”*.
- (3) In past drafts, the first paragraph of the discussion highlighted theories related to chronic stress effects on the hippocampus. We have now removed those references. At present, we instead focus on theories proposing that the hippocampus mediates income-gaps in cognition, and state that our work adds an important distinction to these theories by separately examining the anterior and posterior hippocampus (without mentioning stress).
- (4) In the discussion, we removed a statement about how our findings are consistent with a stress mechanism (which was originally in the second paragraph of the discussion). Thus, we no longer emphasize stress as the core factor when interpreting our results. We also removed the statement about how stress could trigger a cycle of altered stress responses and further structural degradation in the hippocampus. These two changes have the effect of de-emphasizing stress from the conclusions of the manuscript and provide a more balanced perspective.

- (5) In the discussion, we now further emphasize other potential mediating factors. On **page 18, starting on line 354**, we state:

“The large public dataset used here did not include a stress measure, and thus we were unable to directly assess the role that income-related stress played in our findings. Thus, while stress is a plausible mechanism, it is not the only factor that may influence the relations we observed. Indeed, many other factors, including nutrition⁷⁸ and access to material and non-material resources^{2,9,10} correlate with socioeconomic status and may play a role in our findings. For example, relative to families from lower SES backgrounds, higher SES families spend more time engaging children in reading activities^{79,80} and provide more access to educational resources⁵⁵, which are thought to be important for cognitive and neural development⁶. These factors could boost anterior hippocampal volumes⁸¹ and cognitive performance⁵⁵ among individuals from higher income backgrounds, and thus underlie the positive effects of income on cognition and anterior hippocampal volumes that we observed.”

By directly noting the other factors that could have contributed to our findings, stress is further de-emphasized from the discussion.

- (6) In the last draft, the final sentence of the paper focused on how stress could initiate a cycle of hippocampal dysfunction that could lead to an altered stress response. We have now removed this sentence from the paper. Thus, we no longer end on a point about stress, but instead end by motivating future work to better understand the cognitive processes that require more support in low income children.

Remaining concerns regarding analyses of brain volumes:

On page 28, line 577, citation 96 tested ANTS and ASHS, and thus is not an appropriate citation to make the point that adults atlas applies in children sample as the authors are using MAGeT in current study.

We have now removed this citation from the manuscript.

2. On page 31, line 639-641, before combining the left and right volumetric measures, the hemisphere-related effects (e.g., age x hemisphere, SES x hemisphere) in predicting volume should be tested. Only combine if the interactions are not significant.

Thank you for pointing this out. We found that the relationship between hippocampal subregion volumes and age did not differ significantly by hemisphere (all ps > 0.58). Similarly, hemisphere did not moderate the relationship between income and hippocampal subregion volumes (all ps > 0.76).

We now highlight that we tested for these interactions in the text. On **page 29, starting on line 614**, we state: *“After correcting for ICV, we tested whether there were age and hemisphere-related effects on volumes before combining volumes*

across hemispheres. In one set of models, we tested whether hemisphere interacted with age to predict volumes, and in another set of models, whether income interacted with hemisphere to predict volumes (i.e., age x hemisphere; income x hemisphere interactions). We did this separately for the anterior and the posterior hippocampus. We found that the relationship between age and hippocampal subregion volumes did not differ by hemisphere (all ps > 0.58). This suggests that while hippocampal subregion volumes get larger as children get older, this effect doesn't differ significantly by hemisphere. Similarly, we found that hemisphere did not moderate the relationship between income and hippocampal subregion volumes (all ps > 0.76), suggesting that the relationship between income and volumes did not differ across the two hemispheres. Since there were neither age x hemisphere nor income x hemisphere interactions, we calculated bilateral hippocampal volumes by summing analogous regions in the left and right hemispheres. Bilateral hippocampal volumes that had been adjusted for ICV were used in all analyses."

Remaining concern regarding measures of SES: please add limitation of not using income-to-needs ratio in discussion.

We have now added this as a limitation to our discussion section and have called for future researchers to use income to needs ratios in studies investigating how income influences brain and cognitive development. On **page 19, lines 368-372**, we state: *"Moreover, although not available in this dataset, future work investigating the effects of income should consider using an income-to-needs ratio (i.e., income divided by the national poverty threshold for a family of the same size), which would provide a more precise estimate of the amount of resources available to children than income alone."*

Remaining concern regarding treating age: given the large age range non-linear trends need to be tested consistently. Handling of age is still not sufficiently described and non-linear terms are only adopted in one analysis (described starting in line 698), yet, due to the large age range (and good theoretical, and practical reasons) should be adopted consistently.

Thank you for this comment. To be clear, we tested whether the relationship between age and hippocampal volumes followed a nonlinear relationship but did not do the same for the cognitive scores. Below, we explain our rationale. We have also reported results from analyses that include nonlinear age transformations in the supplement for completeness (See **page 52-54, starting on line 1181**).

We tested for a nonlinear relationship between age and hippocampal volumes because it has been reported in prior work (Schlichting, Guarino, Schapiro, Turk-Browne, 2016; Daugherty, Flinn, Ofen, 2017). Because only the linear age term significantly explained variance in hippocampal volumes, we retained the linear

age term only in models for which hippocampal subregions volumes were the dependent variable (See **page 32, starting on line 677**).

In contrast, for models in which cognitive scores were the dependent variable, we did not include nonlinear age terms as covariates. We made this principled decision because we wanted to test whether the hippocampus correlated with cognitive scores above and beyond experience (age). The linear effect of age is a proxy for experience in years, which is inherently linear (unlike hippocampal volumes that may increase and decrease overtime). In contrast, nonlinear age terms are mathematical transformations of age that could very likely be driven by variance in the brain across individuals. Given we had a direct measure of the brain – the hippocampus – and given our question specifically concerned how the hippocampus explains variance in cognitive scores, we thought it was theoretically inappropriate to add nonlinear age transformations to our models that also include brain data to explain cognition. We have now described this rationale in the paper on **page 34, lines 708-716**, and report results from models that include nonlinear age transformations in the supplement (**page 52-54, lines 1181-1219**).

Per your suggestion, we did perform analyses to assess whether nonlinear transformations of age explained variance in cognitive scores. We found that quadratic and cubic transformations of age contributed to explaining variance in memory scores in the full sample. In contrast, within the lower income sample, only the quadratic transformation of age explained significant variance in memory scores. Similarly, only the quadratic age transformation explained significant variance in vocabulary scores in the full sample and low-income subsample.

We have now re-run models with the relevant non-linear age transformations as covariates. We found that all relationship patterns were maintained, but one became marginally significant after adding these nonlinear age transformations. *Income* significantly correlates with episodic memory and vocabulary in both the full sample and lower income subsample, all $ps < 0.05$. *Anterior hippocampal volumes* significantly correlate with memory and vocabulary scores in the full sample and the lower income subsample, all $ps < 0.05$. *Anterior hippocampal volumes* marginally mediated income-gaps in memory scores in the full, $p = 0.096$, and lower income subsample, $p = 0.07$. Lastly, anterior hippocampal volumes significantly mediated income gaps in vocabulary scores in the full, $p = 0.007$, and lower income subsample, $p = 0.03$.

Minor:

1. *When reporting results, explain what r value is for the first time (Page 7, line 110).*

Thank you. On **page 7, lines 114-115**, we now state: “*Of note, the r value refers to the correlation between the two measures described (here, income and cognitive scores) after controlling for age and sex.*”

2. On page 8, line 146, figures 1e & f are mislabeled and should be f&g.

Thank you. We've now corrected this.

3. On page 9, line 158, supplementary tables 55-58 numbers are mislabeled and should be 49-54.

Thank you. We have now corrected references to these tables.

4. For figure 4, add explanation for the value in parentheses.

We have now added an explanation. On **page 17 lines 332-335**, we state: “*The values in parentheses are the standardized beta coefficients reflecting the relationship between the two variables before accounting for anterior hippocampal volumes (i.e., the “total effect”). The values in front of the parentheses reflect the relationship between the variables after accounting for anterior hippocampal volumes (i.e., the “direct” effect).*”

Supplementary Table 36, was the regions (anterior/posterior) entered as repeated measures?

Yes. Because we had two observations per participant, we used a linear mixed effects model rather than a regression model to account for the random effect of participant. Since regions were nested within participants, we modelled random intercepts for participants to account for the random effect of participant on regional volumes. We were unable to model random subregion slopes for each participant since we only have one instance of each subregion per participant. We have now added this information to the manuscript under **Supplementary Table 36** on **page 71-72, lines 1466**.

Reviewer #3 (Remarks to the Author):

I find the authors' response thoughtful and their points valid. I have no further concerns.

***REVIEWERS' COMMENTS:

Reviewer #2 (Remarks to the Author):

The authors have adequately responded to my remaining concerns. I believe that the revisions resulted in a stronger manuscript that more accurately presents their interesting findings.

REVIEWERS' COMMENTS

Reviewer #2 (Remarks to the Author):

The authors have adequately responded to my remaining concerns. I believe that the revisions resulted in a stronger manuscript that more accurately presents their interesting findings.

We thank the review for all of their helpful feedback. We are especially grateful for their time and attention to detail and think their contributions have strengthened the manuscript.